



**Contrasting effects of secondary organic aerosol formations on organic aerosol**
**hygroscopicity**
**Ye Kuang[1,2], Shan Huang[1,2*], Biao Xue[1,2], Biao Luo[1,2], Qicong Song[1,2], Wei Chen[3], Weiwei Hu[3],**
**Wei Li[1,2], Pusheng Zhao[4], Mingfu Cai[1,2], Yuwen Peng[1,2], Jipeng Qi[1,2], Tiange Li[1,2], Duohong**
**Chen[5], Dingli Yue[5], Bin Yuan[1,2], Min Shao[1,2*]**
[1] Institute for Environmental and Climate Research, Jinan University, Guangzhou, China.
[2] Guangdong-Hongkong-Macau Joint Laboratory of Collaborative Innovation for Environmental
Quality, Guangzhou, China.
[3] State Key Laboratory of Organic Geochemistry and Guangdong Key Laboratory of Environmental
Protection and Resources Utilization, Guangzhou Institute of Geochemistry, Chinese Academy of
Sciences, Guangzhou 510640, China
[4] Institute of Urban Meteorology, China Meteorological Administration, Beijing 100089, China
[5] Guangdong Ecological and Environmental Monitoring Center, State Environmental Protection Key
Laboratory of Regional Air Quality Monitoring, Guangzhou 510308, China
*Correspondence to: Shan Huang (shanhuang_eci@jnu.edu.cn) and Min Shao (mshao@pku.edu.cn)
**Abstract**
Water uptake abilities of organic aerosol under sub-saturated conditions play critical roles in direct
aerosol radiative effects and atmospheric chemistry, however, field characterizations of organic aerosol
hygroscopicity parameter $\kappa_{OA}$ under sub-saturated conditions remain limited. In this study, a field
campaign was conducted to characterize $\kappa_{OA}$ at relative humidity of 80% with hourly time resolution
for the first time in the Pearl River Delta region of China. Observation results show that during this
campaign secondary organic aerosol (SOA) dominated total organic aerosol mass (mass fraction >70%
on average), which provides us a unique opportunity to investigate influences of SOA formation on
$\kappa_{OA}$. Results demonstrate that the commonly used organic aerosol oxidation level parameter O/C was
weakly correlated with $\kappa_{OA}$ and failed in describing the variations of $\kappa_{OA}$. However, the variations
in $\kappa_{OA}$ were well reproduced by mass fractions of organic aerosol factor resolved based on aerosol
mass spectrometer measurements. The more oxygenated organic aerosol (MOOA) factor, exhibiting
the highest average O/C (~1) among all organic aerosol factors, was the most important factor driving
the increase of $\kappa_{OA}$ and was commonly associated with regional air masses. The less oxygenated


organic aerosol (LOOA, average O/C of 0.72) factor, revealed strong daytime production, exerting
negative effects on $\kappa_{OA}$. Surprisingly, the aged biomass burning organic aerosol (aBBOA) factor also
formed quickly during daytime and shared a similar diurnal pattern with LOOA, but had much lower
O/C (0.39) and had positive effects on $\kappa_{OA}$. The correlation coefficient between $\kappa_{OA}$ and mass
fractions of aBBOA and MOOA in total organic aerosol mass reached above 0.8. The contrasting
effects of LOOA and aBBOA formation on $\kappa_{OA}$ demonstrates that volatile organic compound (VOC)
precursors from diverse sources and different SOA formation processes may result in SOA with
different chemical composition, functional properties as well as microphysical structure, consequently,
exert distinct influences on $\kappa_{OA}$ and render single oxidation level parameters (such as O/C) unable to
capture those differences. Aside from that, distinct effects of aBBOA on $\kappa_{OA}$ was observed during
different episodes, suggesting that the hygroscopicity of SOA associated with similar sources might
also differ much under different emission and atmospheric conditions. Overall, these results highlight
that it is imperative to conduct more researches on $\kappa_{OA}$ characterization under different
meteorological and source conditions, and examine its relationship with VOC precursor profiles and
formation pathways to formulate a better characterization and develop more appropriate
parameterization approaches in chemical and climate models.


## 1 Introduction

Organic aerosol (OA) composed of hundreds to thousands of organic species is one of the

dominant aerosol components in the atmosphere and exert significant effects on climate and
environment (Jimenez et al., 2009). The water uptake ability of atmospheric organic aerosol plays key
roles in aerosol direct radiative effects and aerosol-cloud interactions (Rastak et al., 2017;Liu and
Wang, 2010), and also aerosol liquid water content (Li et al., 2019;Jin et al., 2020) thus atmospheric
chemistry. However, the hygroscopicity parameter $\kappa_{OA}$ that describes the water uptake abilities of
organic aerosol remains poorly quantified and mechanisms behind $\kappa_{OA}$ variations are not well
understood (Kuang et al., 2020b). Atmospheric OA is usually composed of both primary or secondary
organic aerosol components. Primary OA (POA) is directly emitted from anthropogenic and natural
sources such as biomass burning, coal and fossil fuel combustion, cooking and biogenic emissions.



Whereas secondary OA (SOA) is typically formed through atmospheric oxidation of volatile organic
compounds (VOCs) or aging processes of POA. It is commonly thought that OA becomes more
oxidized during its evolvement in the atmosphere and will in general be more hygroscopic after aging
processes (Jimenez et al., 2009). A few studies have investigated the relationship between $\kappa_{OA}$ and
aerosol oxidation state parameters such as O/C ratio or f44 (fraction of m/z 44 in OA measurements of
aerosol mass spectrometers). Some results, especially those from laboratory studies, demonstrated that
$\kappa_{OA}$ was highly correlated with O/C (Jimenez et al., 2009;Massoli et al., 2010;Kuang et al.,
2020a;Zhao et al., 2016;Lambe et al., 2011), however, other researches demonstrated that $\kappa_{OA}$ was
not or only weakly correlated with O/C (Cerully et al., 2015;Lathem et al., 2013;Yeung et al.,
2014;Alfarra et al., 2013). As the research continues, it was revealed that many factors can have
significant impacts on $\kappa_{OA}$, such as different functional groups, carbon chain length and aerosol liquid
water content, etc. (Rickards et al., 2013;Suda et al., 2014;Petters et al., 2017;Marsh et al., 2017;Liu
et al., 2018). Kuang et al. (2020b) recently reviewed laboratory and field measurements of $\kappa_{OA}$ and
concluded that O/C is not enough in parameterizing $\kappa_{OA}$ and that additional parameters are needed.
Therefore, it is worthwhile and imperative to endeavor on $\kappa_{OA}$ quantifications and parametrizations,
especially, considering that organic aerosol might play more critical roles in atmospheric environment
and climate for decades to come under strict control on anthropogenic emissions.

Most previous studies on $\kappa_{OA}$ focused on laboratory studies, usually investigating $\kappa_{OA}$ of SOA

produced from laboratory chamber systems, which might be far different from real atmospheric SOA
spectral. Quantifications of $\kappa_{OA}$ based on field measurements remain relatively limited and are also
urgently needed to yield complementary information, which in turn might provide guidance for the
design of future laboratory studies. It is important to conduct more researches on $\kappa_{OA}$ spatiotemporal
distributions and examine its relationship with OA profiles to reach a better characterization and give
rise to more appropriate parameterization approaches in chemical and climate models. China is a
country that has been experiencing severe aerosol pollution and has been undergoing rapid changes
under drastic air pollution control measures. However, despite the importance of organic aerosol
hygroscopicity, only few studies attempted to quantify $\kappa_{OA}$ based on field measurements (Wu et al.,
2016;Li et al., 2019;Hong et al., 2018;Gunthe et al., 2011), mainly focusing on the North China Plain
(NCP). The Pearl River Delta (PRD) region is much cleaner than the NCP in terms of particulate matter
pollution, suggesting that distinct regions in China are at different stages of air pollution controls (Xu





et al., 2020). The composition of PM$_{2.5}$ (particulate matter with aerodynamic diameter less than 2.5
um) also differs much among regions, for example, OA and SOA fractions are much higher in the PRD
than those in the NCP and their precursors are also much different (Zhou et al., 2020a). More
investigations on $\kappa_{OA}$ based on field studies in regions other than the NCP are urgently required.

In addition, most field studies on $\kappa_{OA}$ only gave an estimate of the average $\kappa_{OA}$ (Gunthe et

al., 2011) or an average statistical relationship between $\kappa_{OA}$ and O/C (Wu et al., 2013) and only few
studies have reported $\kappa_{OA}$ of higher time resolution featuring its diurnal variation characteristics
(Deng et al., 2019), and almost no studies have reported $\kappa_{OA}$ with high time resolution. Kuang et al.
(2020a) proposed a new method to estimate $\kappa_{OA}$ based on aerosol optical hygroscopicity
measurements and bulk aerosol chemical composition measurements, which yielded $\kappa_{OA}$ estimates
at hourly time resolution. Based on this dataset, it was found that variations in $\kappa_{OA}$ were highly
correlated with mass fractions of oxygenated organic aerosol in OA. In this study, the same method
was applied to the dataset acquired from field measurements at a background site of the PRD region.
High time resolution characterization of $\kappa_{OA}$ and aerosol chemical properties were also achieved,
which enabled us to dig deeper on what factors other than O/C drove the variations of $\kappa_{OA}$ and to
further elucidate on the complexity and possible approaches in parameterizing $\kappa_{OA}$ based on field
measurements. Details on aerosol measurements and the $\kappa_{OA}$ estimation method were presented in
Sect.2 and Sect.3, respectively. An overview of campaign data and general factors driving aerosol
chemistry was presented in Sect 4.1. The variations in estimated $\kappa_{OA}$ and its relationship to OA
oxidation state and to distinct OA factors were presented and discussed in Sect 4.2. The complexity
regarding $\kappa_{OA}$ was further demonstrated and elucidated in Sect 4.3.
**2 Measurements**
**2.1 Sampling site**
Physical, optical and chemical properties of ambient aerosol particles as well as meteorological
parameters and gas pollutants such as CO, O$_3$ and NO$_x$ were continuously measured during autumn
(from 30$^{th}$ September to 17$^{th}$ November 2018) at a rural site in Heshan county, Guangdong province,
China. This site locates at a small mountain (22°42'N, 112°55'E, altitude of 55 m), about 55 km away
from megacity Guangzhou and is surrounded by villages and small residential towns and thus is little
influenced by local industrial sources. The location of this site is also shown in Fig.S1.



## 2.2 Aerosol physical properties measurements


During this field campaign, instruments were placed in an air-conditioned room. Two inlets were
housed on the roof of the three-floors building for aerosol sampling and both inlets are about 1.8 m
above the floor. One of the inlets was a $PM_{10}$ impactor with a 1.8 m long Nafion drier that lowers the
sample RH down to less than 30% placed downstream of it. A flow splitter was placed below the drier
and instruments downstream of this splitter include an Aerodynamic Particle Sizer (APS, TSI Inc.,
Model 3321, flow rate of 5 L/min), which measured the size distribution of ambient aerosol particles
of aerodynamic diameter about 600 nm to 20 μm; an AE33 aethalometer (Drinovec et al., 2015) with
a flow rate of 5 L/min, which measures aerosol absorption coefficients at seven wavelengths; a
humidified nephelometer system with a flow rate of about 6 L/min. The total flow rate of these
instruments was about 16 L/min, which was quite close to the flow rate of 16.7 L/min required by the
$PM_{10}$ impactor. Thus, these instruments measured physical and optical properties of $PM_{10}$ particles.
The humidified nephelometer system is a laboratory self-assembled one, including two Aurora
3000 nephelometers. One nephelometer measures aerosol scattering properties (scattering and back
scattering coefficients at three wavelengths: 450 nm, 525 nm, 635 nm) at a reference RH (called dry
Nephelometer because of sampling RH is lower than 30%), and another nephelometer measures
aerosol scattering properties under a fixed RH of 80% (called wet Nephelometer and the actual
sampling RH fluctuates within $\pm 1\%$). Details on the humidifier and control algorithm can be found
in Kuang et al. (2020a). Note that to make sure the accuracy of the measured RH in the sensing volume
of the wet Nephelometer, three Vaisala HMP110 sensors with accuracies of $\pm 0.2$ ℃ and $\pm 1.7\%$
for RH between 0 to 90% were used to monitor the RH at different parts of the wet nephelometer. Two
were placed at the inlet and outlet of the wet nephelometer, one was placed in the sensing volume. The
water vapor pressure calculated from these three sensors must be strictly consistent with each other
(relative difference between any two of the sensors must be less than 2 %). Then the sampling RH of
the wet nephelometer was calculated using the averaged water vapor pressure and the sample
temperature measured by the sensor placed in the sensing volume of the wet nephelometer.
Another inlet was connected with a $PM_{2.5}$ impactor (BGI SCC2.354, cut diameter of 2.5 μm with
flow rate of 8 L/min) and was also equipped with a Nafion drier lowering sampling RH down to less
than 30%. Downstream of this inlet were a soot particle aerosol mass spectrometer (SP-AMS,



Aerodyne Research, Inc., Billerica, MA, USA) and a scanning mobility particle sizer (SMPS; TSI
model 3080), which measured particle number size distribution (PNSD) ranging from 10 nm to 760
nm.

### 151   2.3 SP-AMS measurements and data analysis

The SP-AMS was deployed to measure size-resolved chemical composition for submicron
aerosol particles. The SP -AMS is basically a high-resolution time-of-flight aerosol mass spectrometer
(HR-ToF-AMS) combining a laser vaporization device, i.e., soot particle (SP) mode. The instrument
principle has been provided in previous papers (Canagaratna et al., 2007;Onasch et al., 2012). In brief,
HR-ToF-AMS containing a tungsten vaporizer can provide information of those particulate species
vaporized under around 600℃°. By adding a Nd:YAG (1064nm) laser module inside of the HR-ToF-
AMS, the vaporizing temperature can increase to around 4000℃, enabling the SP-AMS to detect
refractory compositions such as black carbon (BC) and metals. After vaporized, the gaseous
components are ionized in electron impact (70eV) way and then quantitatively measured by a time-of-
flight mass spectrometer. Controlled by the orifice as well as aerodynamic lens of SP-AMS, particles
with diameter in submicrometer range are measured. During the Heshan Campaign, SP-AMS was
located next to a SMPS to minimize the sampling discrepancy. The SP-AMS alternately switched
between the V-mode (only tungsten vaporizer) and SP-mode (laser and tungsten vaporizer). The
original time resolution of SP-AMS data was 1 min (per run), and 15min average values were used in
this study to avoid disturbance from mode switching. During the campaign, the calibration system for
SP-AMS was not available and we used the values of ionization efficiency (IE) and relative ionization
efficiency (RIE) of different species from the latest successful calibration. The applied RIEs for default
SP-AMS species are: 1.1 for nitrate, 4 for ammonium, 1.2 for sulfate, 1.4 for organics and 1.3 for
chloride. The composition dependent collection efficiency (CDCE) was applied to mentioned species
following the instruction of Middlebrook et al.(2012). Refractory BC from SP-AMS was calculated
by mass concentration of family $C_x$ ions from high resolution mass spectrometer times a scaling factor
(8) derived by comparison with equivalent BC mass concentration from AE33. SP-AMS data
evaluation was performed by specific software Squirrel (v1.61) for unit mass resolution and Pika
(v1.21) for high resolution based on Igor Pro (v6.37, WaveMetrics, Inc., Oregon, USA). Aside from



the good consistency between the aerosol from derived from SMPS and SP-AMS components as
mentioned in Sect.3.2, the resulting mass concentrations from SP-AMS were further validated by
consistency with the results from external measurements in the same site, e.g., filter measurements and
online measurements using gas aerosol collection system (GAC) with ion chromatography. More
details of SP-AMS data quality assurance will be provided in a parallel paper (Huang et al., in
preparation).
The source apportionment of organic aerosols (OA) was performed by positive matrix
factorization (PMF) based on high resolution OA data collected in V-mode (only tungsten vaporizer).
As a wildly used source analysis method, PMF has been described in previous papers (Paatero,
1997;Paatero and Tapper, 1994). PMF using AMS data can be conducted by an Igor Pro-based panel,
i.e., PMF Evaluation Tool (PET, v2.06, Ulbrich et al., 2009). We input the matrices for OA mass
concentration and uncertainty into the model and operated it according to the instruction in Ulbrich et
al. (Ulbrich et al., 2009). Isotopes and ions with m/z >120 were excluded to minimize the interference
from repeatedly calculated uncertainties and noise signals. In total, 454 ions were considered in PMF.
After investigating different solutions with factor number from 2 to 10, a six-factor solution was chosen
based on the best performance shown by PMF quality parameters and most reasonable source
identification. Two primary OA factors were identified including a hydrocarbon-like OA (HOA,
containing cooking emissions) and a biomass burning OA (BBOA). The other four factors were related
to secondary formation or ageing process: 1) more oxygenated OA (MOOA, regional transport), 2)
less oxygenated OA (LOOA, related to daytime photochemical formation), 3) nighttime-formed OA
(Night-OA), and 4) aged BBOA (aBBOA). The mass spectral profile and time series of OA factors
were shown in Figure S3, and OA factors with identified sources will be discussed in Set. 4. More
details on PMF solution selection and source identification will be provided in a parallel paper (Huang
et al., in preparation).

**3 Methodology**
**3.1 Aerosol hygroscopicity derivation from aerosol light scattering measurements**
The aerosol light scattering enhancement factor $f(\text{RH}, \lambda) = \frac{\sigma_{sp}(RH,\ \lambda)}{\sigma_{sp}(dry\ \lambda)}$, $\sigma_{sp}(RH,\ \lambda)$ is the





aerosol scattering coefficient at light wavelength of $\lambda$ and condition of RH, and was only measured
at 80% RH. Thus the aerosol hygroscopicity parameter $\kappa_{f(\mathrm{RH})}$ was derived from $f(80\%, 525\,\mathrm{nm})$.
The principle of this method is to find a diameter independent hygroscopicity parameter $\kappa$ that fits the
observed $f(80\%, 525\,\mathrm{nm})$ best. Although Kuang et al. (2017) proposed a simple method for
deriving $\kappa_{f(\mathrm{RH})}$ based only on measurements of the humidified nephelometer system, in this study,
the more traditional method described therein was adopted to retrieve $\kappa_{f(\mathrm{RH})}$, which uses
measurements of PNSD as inputs of Mie theory and the $\kappa$-Köhler theory. The idea of deriving $\kappa_{f(\mathrm{RH})}$
from aerosol light scattering measurements was first proposed by Chen et al. (2014), however, the
physical understanding of $\kappa_{f(\mathrm{RH})}$ was not mathematically interpreted until the study of Kuang et al.
(2020a). Briefly, $\kappa_{f(\mathrm{RH})}$ can be approximately understood as the overall hygroscopicity of aerosol
particles with aerosol scattering coefficient contribution as the weighting function for size-resolved $\kappa$
distribution. Results of Kuang et al. (2020a) demonstrated that for typical continental aerosols $\kappa_{f(\mathrm{RH})}$
represents the overall hygroscopicity of aerosol particles with a dry diameter range between 200 and
800 nm, thus no matter if $\kappa_{f(\mathrm{RH})}$ values were retrieved based on aerosol light scattering enhancement
factor measurements downstream of a $PM_{10}$ or a $PM_1$ impactor, they are almost the same, which was
confirmed by direct measurements in Kuang et al. (2020a) (observed average relative difference about

3.5% ).

### 3.2 Organic aerosol hygroscopicity derivation based on aerosol chemical composition and optical hygroscopicity measurements

Aerosol hygroscopicity parameter $\kappa$ were usually calculated using measured aerosol chemical
composition based on volume mixing rule ($\kappa_{chem}$) to represent the aerosol hygroscopicity of aerosol
particles of certain diameters or present the overall hygroscopicity of the entire aerosol populations of
PM1. In this study, the size-resolved aerosol chemical compositions of $PM_1$ were measured using the
SP-AMS, however, the overall aerosol hygroscopicity was only derived based on aerosol light
scattering measurements of $PM_{10}$ bulk aerosols. Results of (Kuang et al., 2020a) demonstrated that
$\kappa_{chem}$ calculated based on bulk chemical compositions of $PM_1$ are quite consistent with $\kappa_{f(\mathrm{RH})}$ of
$PM_1$ therefore also consistent with $\kappa_{f(\mathrm{RH})}$ of $PM_{10}$. We have simulated the $\kappa_{f(\mathrm{RH})}$ of $PM_{10}$ and
$\kappa_{chem}$ of $PM_1$ under different PNSDs coupled with different size-resolved $\kappa$ distribution scenarios, as





shown in Fig.S2a. As shown in the results in Fig.S2b, $\kappa_{f(RH)}$ of PM$_{10}$ and $\kappa_{chem}$ of PM$_1$ are quite
close to each other and the simulated average relative difference ($\frac{\kappa_{f(RH),PM_{10}} - \kappa_{chem,PM_1}}{\kappa_{chem,PM_1}} \times 100\%$) was -
0.4±3%. Thus, $\kappa_{f(RH)}$ of PM$_{10}$ was used as the measured $\kappa_{chem}$ in the following discussions.
The SP-AMS measures size-resolved PM$_1$ mass concentrations of $SO_4^{2-}$, $NO_3^-$, $NH_4^+$, $Cl^-$ and
organic aerosol, thus provides their bulk mass concentrations. A simplified ion pairing scheme was
used to derive mass concentrations of different inorganic salts (as listed in Tab.1) based on measured
bulk ion mass concentrations (Gysel et al., 2007;Wu et al., 2016). Note that the hygroscopicity
parameter was measured at RH of 80%, the κ values of ammonium sulfate and ammonium nitrate at
80% RH were predicted using the Extended Aerosol Inorganic Model (E-AIM), whose predictions for
ammonium nitrate and ammonium sulfate has been proven to be consistent with laboratory results
(Luo et al., 2020;Jing et al., 2018), and those of potassium chloride and ammonium bisulfate were
consistent with Liu et al. (2014)
**Table 1**. Densities (ρ) and hygroscopicity parameters (κ) of inorganic salts used in this study

| Species | $NH_4NO_3$ | $NH_4HSO_4$ | $(NH_4)_2SO_4$ | $KCl$ |
|---|---|---|---|---|
| | (AN) | (ABS) | (AS) | (PC) |
| ρ (g $cm^{-3}$) | 1.72 | 1.78 | 1.769 | 1.98 |
| κ | 0.56 | 0.56 | 0.56 | 0.89 |

Note that $Cl^-$ was coupled with $K^+$ due to that biomass burning events prevailed during this field
campaign. The simple volume mixing rule called Zdanovskii–Stokes–Robinson (ZSR) was usually
used for $\kappa_{chem}$ calculations, that is, bulk $\kappa_{chem}$ of PM$_1$ can be calculated on the basis of volume
fractions of different compounds ($\varepsilon_i$) (Petters and Kreidenweis, 2007) using the following equation:
$\kappa_{chem} = \sum_i \kappa_i \cdot \varepsilon_i$     (1)
And $\kappa_i$ and $\varepsilon_i$ are hygroscopicity parameter κ and volume fraction of compound $i$ in the mixture.
Based on Eq.2 and Tab,1, $\kappa_{chem}$ can be formulated as follows:
$\kappa_{chem} = \kappa_{AS}\varepsilon_{AS} + \kappa_{AN}\varepsilon_{AN} + \kappa_{ABS}\varepsilon_{ABS} + \kappa_{PC}\varepsilon_{PC} + \kappa_{BC}\varepsilon_{BC} + \kappa_{OA}\varepsilon_{OA} + \kappa_X\varepsilon_X$     (2)
where $\kappa_{OA}$ and $\varepsilon_{OA}$ are κ and volume fraction of entire organic aerosol populations,$\kappa_X$ and $\varepsilon_X$ are
κ and volume fraction of aerosol constituents which are beyond the detection ability of the SP-AMS.
The hygroscopicity of these unidentified aerosol species, in continental regions, likely be dust but still



possible composed of other components such as biogenic primary aerosol, were not discussed before.
On the basis of current literature reports, dust is nearly hydrophilic and varies a lot, with $\kappa$ of mineral
dust and road dust as well as oil or coal fly ash are in the range of 0.01 to 0.08 (Koehler et al., 2009;Peng
et al., 2020). In this paper, $\kappa_X$ is arbitrarily specified as 0.05. The $\varepsilon_X$ are estimated as the $PM_1$
volume concentration difference between measured by the SMPS and calculated from volume
concentration summation of chemical compounds listed in Tab.1 and volume concentrations of BC
and organic aerosol, and the estimated average contribution $\varepsilon_X$ during this campaign is 13% as shown
in Fig.S4. In the volume concentration calculations of BC and organic aerosol, BC density of 1.7 $g/cm^3$
was assumed, and organic aerosol density is calculated based on the density parameterization shame
proposed by Kuwata et al. (2012) using the organic aerosol elemental ratios O:C and H:C measured
by the SP-AMS as input parameters. In addition, $\kappa_{BC}$ was set to zero due to the hydrophilic property
of BC particles. Then, $\kappa_{OA}$ can be estimated based on measured $\boldsymbol{\kappa_{chem}}$ using the following formula:
$$\kappa_{OA} = \frac{\kappa_{chem} - (\kappa_{AS}\varepsilon_{AS} + \kappa_{AN}\varepsilon_{AN} + \kappa_{ABS}\varepsilon_{ABS} + \kappa_{PC}\varepsilon_{PC} + \kappa_X\varepsilon_X)}{\varepsilon_{OA}} \qquad (3)$$
**4 Results and discussions**
**4.1 Overview of the campaign data**

The time series of meteorological parameters such as wind speed, wind direction, RH and ambient

air temperature, aerosol scattering coefficients, aerosol hygroscopicity parameter $\kappa_{f(RH)}$, mass
concentrations of aerosol components as well as gas pollutant concentrations are shown in Fig.1.
During this campaign, the RH mainly ranged from 50% to 80% with an average (± 1σ) of 60±14%,
with the nighttime RH frequently reached beyond 70%, which favors the nighttime aqueous phase
chemistry. Temperatures mainly ranged from 18 to 28 ℃, with an average (± 1σ) of 23.6±3.3 ℃,
indicating a relatively warm state during this campaign though in the autumn. The aerosol scattering
coefficients at 525 nm ($\sigma_{sp,525}$) shown in Fig.1b demonstrate $\sigma_{sp,525}$ generally ranged between 20 to
600 $Mm^{-1}$, with an average (± 1σ) of 256±102 $Mm^{-1}$, indicating moderately polluted conditions during
this campaign. The NR-$PM_1$ mass concentrations ranged from 1 to 94 $\mu g/m^3$, with an average (± 1σ)
of 33±14 $\mu g/m^3$. Nitrate, sulfate, ammonium and organic aerosol contributed on average 19%, 11%,



9% and 58% to total NR-PM$_1$, which was consistent with the aerosol chemical compositions typically
observed in the PRD region featuring organic aerosol as the major constituent of NR-PM$_1$ and higher

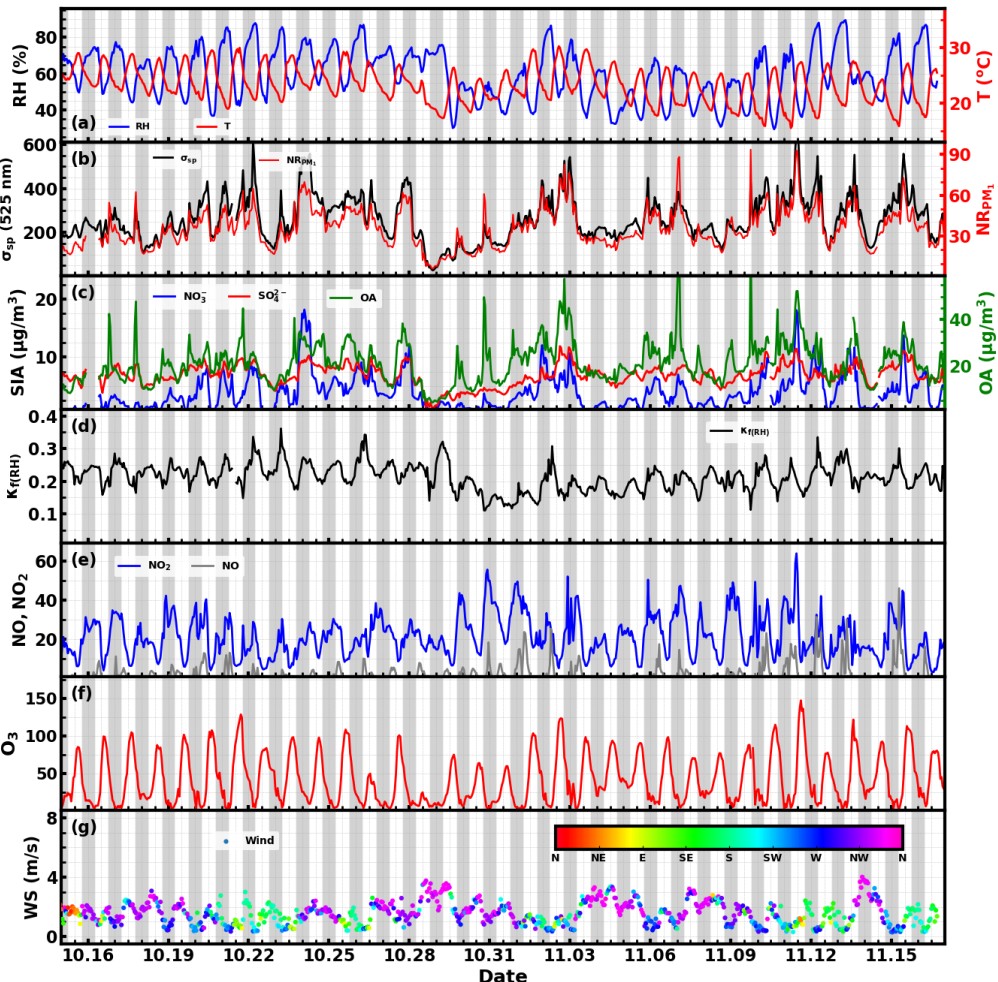

**Figure 1.** Time series of **(a)** RH and temperature; **(b)** aerosol scattering coefficient at 525 nm and mass concentrations of PM$_1$ non-refractory components; **(c)** mass concentrations of sulfate, nitrate and organic aerosol; **(d)**The hygroscopicity parameter $\kappa$ retrieved from aerosol light scattering enhancement measurements; **(e)** NO and NO$_2$ concentrations; **(f)** O$_3$ concentration; **(g)** wind speed/direction. Shaded gray areas corresponding to nighttime periods.

sulfate concentration than nitrate concentration (Zhou et al., 2020b). However, the NR-PM$_1$
composition profile differed much from those recently observed in urban Guangzhou (Guo et al., 2020),
a megacity about 100 km away from Heshan, where sulfate concentrations were on average only



slightly higher than nitrate concentrations during autumn and winter seasons of 2017. The large mass
contribution of organic aerosol in $PM_1$ resulted in generally moderate ambient aerosol hygroscopicity,
with $\kappa_{f(RH)}$ ranging between 0.11 and 0.36 with an average (± 1σ) of 0.22±0.04. The small standard
deviation further suggests for relatively small variations in aerosol hygroscopicity. Sulfate
concentrations showed much less daily and diurnal variations than those of nitrate and organic aerosol,
suggesting that the sulfate level was determined by the regional scale background, while nitrate and
organic aerosol concentration were significantly influenced by local production. Especially, the nitrate
concentration usually experienced a sharp increase since sunset and peaks after mid night, sometimes
even reached beyond sulfate mass concentration. The time series of $NO_2$, NO and $O_3$ concentration are
also shown in Fig.1e and Fig.1f. $NO_2$ concentration showed distinct diurnal variations, and ranged
from 3.5 to 64 ppb with an average (± 1σ) of 20.5±10.5 ppb. The NO concentration ranged from
almost 0 to 45 ppb with an average (± 1σ) of 2.2±4.5 ppb, indicating generally low concentrations of
NO. $O_3$ concentrations ranged from 2 to 147 ppb with an average (± 1σ) of 41.5±31.4 ppb, frequently
reaching over 90 ppb in the afternoon, indicating for strong daytime photochemistry, and dropped
rapidly after sunset towards a very low concentration (usually below 5 ppb) after midnight.

The average diurnal variations of $NO_2$, NO, $O_3$, CO, aerosol chemical compositions, $\kappa_{f(RH)}$ and

meteorological parameters are shown in Fig.2. $O_3$ concentrations began to increase after sunrise,
peaked near 15:00 and then began to decrease quickly but drops slower after midnight. Meanwhile,
NO concentration began to decrease quickly after sunrise, reached and remained near zero after
noontime, and began to slightly increase after 21:00. $NO_2$ concentration increased quickly after 15:00
and reached a plateau after 21:00. Variation characteristics of NO, $O_3$, and $NO_2$ suggest that the
relatively low NO concentration resulted in weak titration effects on $O_3$, where upon typical $NO_3$
chemistry and subsequent $N_2O_5$ chemistry was likely to occur, which was likely the mechanism behind
the observed nitrate variations. Nitrate concentrations increased quickly since 16:00 and peaked after
midnight (about 03:00 LT), further confirming this speculation.

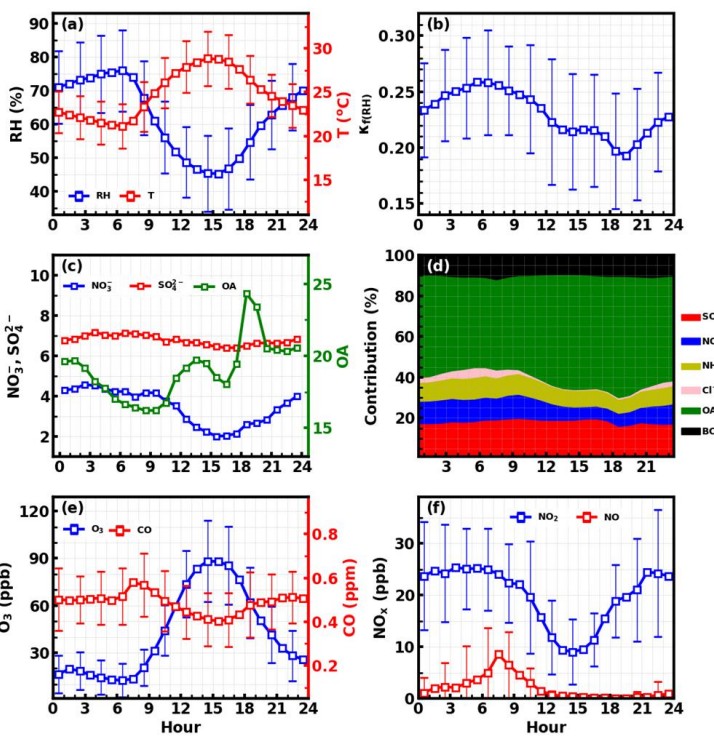

**Figure 2.** Average diurnal variations of **(a)** RH and T; **(b)** $\kappa_{f(RH)}$; **(c)** sulfate, nitrate and organic aerosol; **(d)** mass fractions of different components; **(e)** $O_3$ and CO; **(f)** $NO_2$ and NO.

Under the strong daytime photochemistry and nighttime heterogenous formation of nitrate,
evident diurnal variations of aerosol hygroscopicity was observed. The overall aerosol hygroscopicity
variation was generally consistent with the variation pattern of inorganic aerosol fraction in NR-PM$_1$
as shown in Fig.2d. In detail, the overall variations of nitrate and associated ammonium, as well as
organic aerosol determines the general hygroscopicity variation pattern: the quick increase in organic
aerosol between 16:00 to 19:00 resulted in the quick $\kappa_{f(RH)}$ decrease during this period; since then
the general decrease of organic aerosol and increase of nitrate resulted in the increase of $\kappa_{f(RH)}$ until
the next morning; the daytime decrease of nitrate and increase of organic aerosol resulted in a $\kappa_{f(RH)}$
decrease before 13:00. Note that sulfate concentration remaining almost constant throughout the day
further confirmed previous statement that local production likely contributed less to sulfate





concentration, which can be an indicator of regional air mass status.

These results suggest that both strong daytime photochemistry and nighttime NO₃ chemistry

played significant roles in diurnal variations of organic aerosol and nitrate, while aged regional air
mass determined the sulfate concentration, which provides a good opportunity for investigating how
typical daytime photochemistry and nighttime NO₃ chemistry and aged regional organic aerosol
components impact on organic aerosol hygroscopicity.
**4.2 $\kappa_{OA}$ derivations and its relationship with organic aerosol oxidation state**

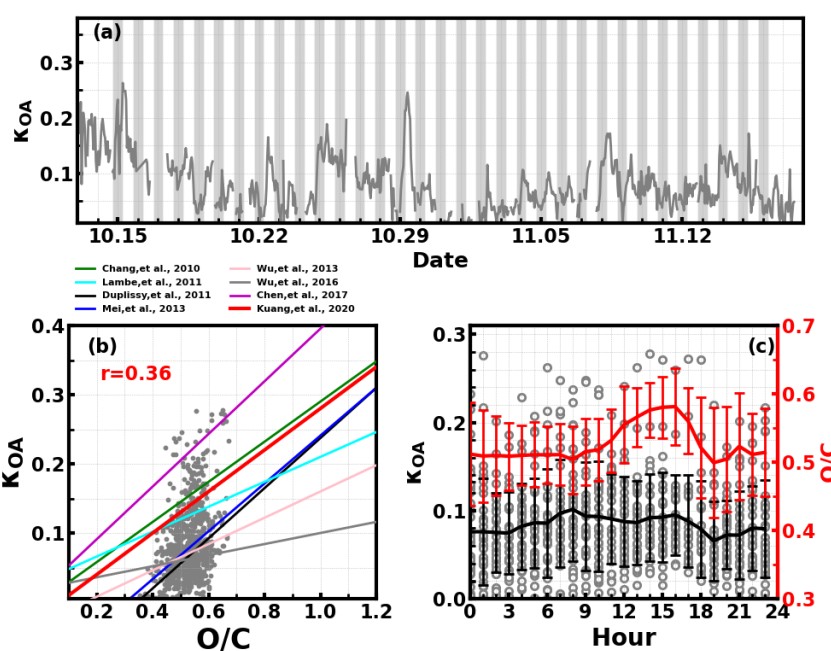

**Figure 3.** **(a)** Time series of derived $\kappa_{OA}$; **(b)** Correlations between O/C ratio and $\kappa_{OA}$, lines correspond to empirical relationships between $\kappa_{OA}$ and O/C ratio reported in different studies; **(c)** Diurnal variations of $\kappa_{OA}$ and O/C ratio;

The organic aerosol hygroscopicity parameter $\kappa_{OA}$ was derived according to the method

mentioned in Sect.3.2, and the results with hourly time resolution are shown in Fig.3a. $\kappa_{OA}$ revealed
daily and diurnal variations, and ranged from almost zero to 0.28 with an average (± 1σ) of 0.085±0.05.
The relationship between $\kappa_{OA}$ and O/C was further investigated and shown in Fig.3b. Results



demonstrated that $\kappa_{OA}$ and O/C were weakly correlated during this campaign, with most data points
falling in the published $\kappa_{OA}$ and O/C relationship band. During this campaign, O/C generally resided
in a small range (from about 0.4 to 0.6) with an average (± 1σ) of 0.053±0.006, indicating small
variations in O/C, however, featuring drastic variations in $\kappa_{OA}$. The average diurnal variations of O/C
and $\kappa_{OA}$ are shown in Fig.3c. On average, $\kappa_{OA}$ increased slowly during the nighttime and varied
even smaller during most of the daytime. Nevertheless, it experienced a relatively quicker decrease
from 17:00 to 19:00, which appeared to be coincident with the quick OA concentration increase as
shown in Fig.2. However, the O/C increased during the period when O₃ concentration increased
quickly, suggesting that daytime photochemistry drove the OA oxidation during daytime. The key
point here is that the diurnal patterns of O/C and $\kappa_{OA}$ differed much from each other, which is why

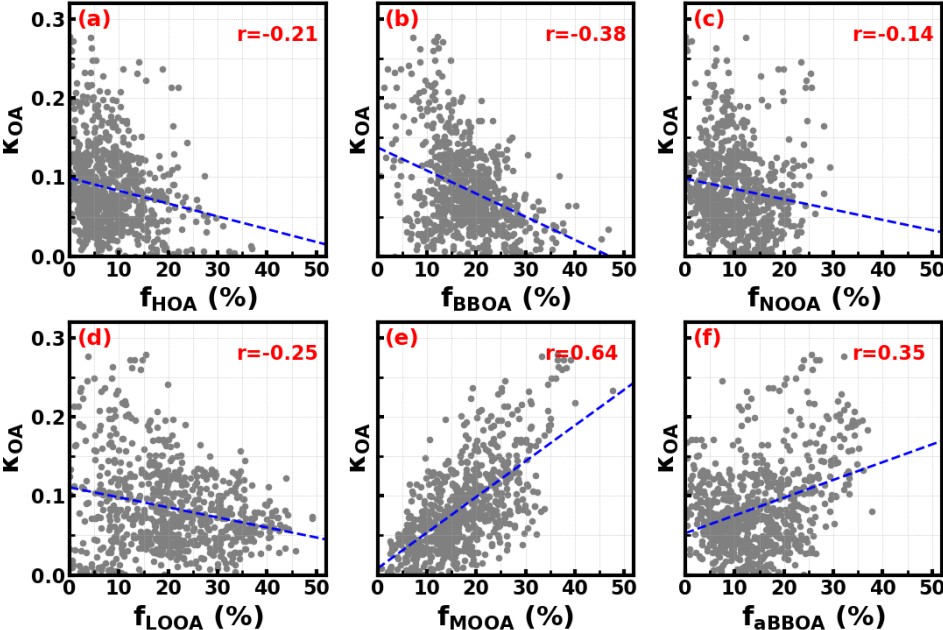

**Figure 4**. Correlations between $\kappa_{OA}$ and mass fractions of OA factors in total OA mass.

the variation in O/C failed to describe that of $\kappa_{OA}$.

The question remains which factors were controlling the variations of $\kappa_{OA}$. The relationships

between $\kappa_{OA}$ and mass fractions of different PMF OA factors in total OA mass were further
investigated and shown in Fig.4. In general, the average (± 1σ) mass fractions of HOA, BBOA, aBBOA,
LOOA, Night-OA, and MOOA were: 8.7% (± 7.8%), 16.5% (± 8.3%), 15.9% (± 10.5%), 19.1% (±




10.9%), 10.4% (± 6.5%), 18.6% (± 12.2%), and it means that during this campaign SOA dominates
organic aerosol (SOA > 70%). Two primary OA factors, HOA and BBOA were related to vehicle
exhausts mixed with cooking emissions and to biomass burning emissions, respectively. $\kappa_{OA}$ was
negatively correlated with both HOA and BBOA, which is consistent with previous literature reports
that primary OA components such as HOA and BBOA are generally hydrophobic. The average diurnal
variations of OA PMF factors shown in Fig.5 demonstrate that both BBOA and HOA peaked near
18:00, which should be associated with the frequently observed biomass burning events and supper
cooking in villages near the site. This explained the sharp increase of OA mass and the sharp decrease
near 18:00 as shown in Fig.3c. It was generally thought that secondary aerosol formation would result

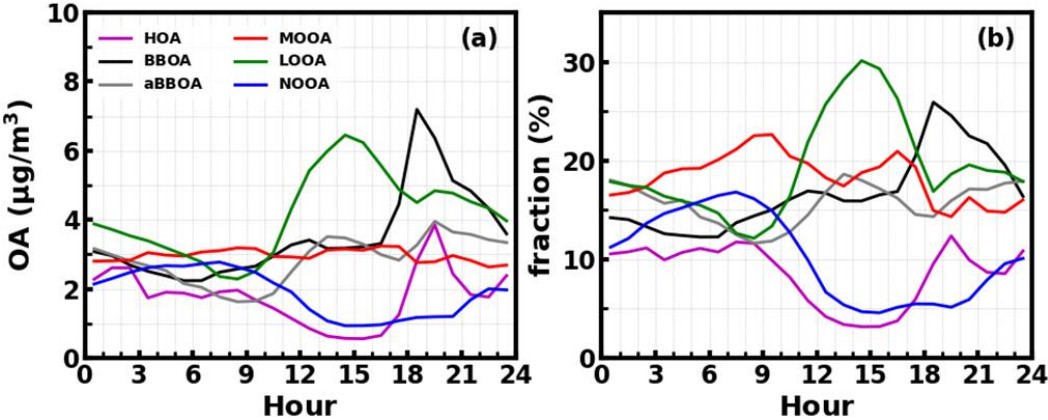

**Figure 5**. Average diurnal variations of mass concentrations **(a)** and their mass fractions **(b)** in total OA mass of different PMF OA factors.

in increases of aerosol hygroscopicity. However, $\kappa_{OA}$ was also negatively correlated with LOOA
(Fig.4d), whose mass concentration increase rapidly after sunrise and are likely secondary due to local
photochemistry with potential precursors such as isoprene and anthropogenic VOCs. The average O/C
ratio for LOOA is 0.72, which is only lower than that of MOOA, suggesting that the daytime OOA
formation and decrease of BBOA and HOA mass concentrations drove the increase of daytime O/C,
and the negative correlation between $\kappa_{OA}$ and LOOA mass fraction explained why O/C failed to
describe diurnal variations of $\kappa_{OA}$.   $\kappa_{OA}$ was also negatively correlated with Night-OA fraction,
which increased during nighttime and was highly correlated with nitrate concentrations (Figure S4),
which were likely associated the $NO_3$ nighttime chemistry as discussed in Sect. 4.1. Results of Suda
et al. (2014) demonstrated that the addition of $NO_3$ radical would exert negative impacts on $\kappa_{OA}$,



which is consistent with the observations shown here. As shown in Fig.4, $\kappa_{OA}$ was positively
correlated with both MOOA and aBBOA, especially with that of MOOA. MOOA was highly
correlated with sulfate and showed almost no diurnal variations, indicating that the highly oxygenated
(O/C ~1) MOOA was also more associated with regional air masses. The observed small nighttime
increase of $\kappa_{OA}$ could be associated with the slight increase in MOOA mass fraction as shown in
Fig.5b. Similar to LOOA, the aBBOA increased during daytime, which revealed quick ageing process
of biomass burning related precursors or primary aerosols through photochemistry. Also, the aBBOA
factor showed similar variation trend with $C_6H_2NO_4^+$ (m/z 151.998, see Fig. S3) which is a
characteristic ion of a typical aged BBOA component nitrocatechol (Bertrand et al., 2018). However,
the resolved average O/C ratio of aBBOA was only 0.39, which is even lower than that of BBOA (O/C
~ 0.48), implying that BBOA were likely formed through oxidation of gaseous BBOA precursors rather
than the direct oxidation of BBOA. The fact that nitrocatechol is more likely to be contributed by
oxidation of gaseous precursors in biomass burning plumes rather than primary biomass burning
emissions (Wang et al., 2019) rationalizes this speculation. The similar diurnal characteristics but
contrasting effects of LOOA and aBBOA on $\kappa_{OA}$ further explains the weak correlation coefficient
between $\kappa_{OA}$ and O/C. However,the weak but positive correlation between $\kappa_{OA}$ and O/C should
have arose from the much stronger positive correlation between $\kappa_{OA}$ and MOOA mass fractions.
LOOA has relatively high O/C and its abundance usually reaches above that of MOOA during the
afternoon, however, its negative effects on $\kappa_{OA}$ was partially compensated by aBBOA which had
lower O/C. In addition, $\kappa_{OA}$ was mostly associated with mass fractions of MOOA with highest O/C,
thus giving rise to the weak but positive relationship between $\kappa_{OA}$ and O/C. As for $\kappa_{OA}$ diurnal
variations, daytime increase of aBBOA and LOOA has compensating effects on $\kappa_{OA}$, and the HOA
and Night-OA decrease further complicated its variations.
**4.3 Discussions on complexity of organic aerosol hygroscopicity parameterizations**
As demonstrated in Sect.4.2, the LOOA factor with higher O/C had negative impacts on $\kappa_{OA}$,
while aBBOA with much lower O/C had positive effects on $\kappa_{OA}$. These results suggested that O/C is
not enough for parameterizing $\kappa_{OA}$ and the question remains what additional parameters are needed
or how should they be implemented? To further explore on this issue, the relationships between $\kappa_{OA}$





and mass fractions of aBBOA+MOOA in total OA mass ($f_{MOOA+aBBOA}$) was further investigated to
manifest the complexity of $\kappa_{OA}$ variations and discuss potential impact factors, with results shown in
Fig.6a. As discussed in Sect.4.2, both MOOA and aBBOA had positive effects on $\kappa_{OA}$, however, the
relationship between $\kappa_{OA}$ and $f_{MOOA+aBBOA}$ does not yield a higher correlation coefficient than that
between $\kappa_{OA}$ and $f_{MOOA}$, and the results shown in Fig.6a demonstrate that $\kappa_{OA}$ and $f_{MOOA+aBBOA}$
might have different relationships during different periods. The relationships between $\kappa_{OA}$ and
$f_{MOOA+aBBOA}$ during three periods were further investigated and shown in Fig.6b-d, which shows that
during the first period from 10-12 to 10-22, $\kappa_{OA}$ was highly correlated with $f_{MOOA+aBBOA}$ (R=0.82),
with all points falling in a narrow band, suggesting that $f_{MOOA+aBBOA}$ alone could describe the variations
in $\kappa_{OA}$ well. However, during the second period (from 10-23 to 11-02) and the third period (from 11-
03 to 11-17) the correlation coefficients between $\kappa_{OA}$ and $f_{MOOA+aBBOA}$ were much lower. Obviously,
$f_{MOOA+aBBOA}$ during the second and the third period was in general much lower than that during the
first period. The timeseries of $\kappa_{OA}$ and different PMF OA factors are shown in Fig.7. MOOA
displayed relatively small variations during this campaign, highlighting that the regional air mass did
not experience tremendous variations, and suggesting that changes of other OA factors especially
aBBOA have resulted in different relationships between $\kappa_{OA}$ and $f_{MOOA+aBBOA}$. The results in Fig.7c
shows that the ratio between aBBOA and BBOA differs much during three periods and declines from
the first period to the third period. During the first period, aBBOA was more abundant and was well
correlated (R= 0.57) with BBOA. At the same time, aBBOA was positively correlated with HOA (R =
0.49) especially with the cooking emission tracer $C_6H_{10}O^+$ (R = 0.60), which could be emitted together
with biomass burning emissions, when residents in surrounding villages cooked with biomass fuels.
BBOA and aBBOA had comparable levels during the second period, however, aBBOA concentration





was much lower than that of BBOA during the third period. It can also be noticed that aBBOA in the
second period showed higher correlation with BBOA (R = 0.45) than that in the last period (R = 0.17),
which was also the case with cooking emission tracer (R = 0.60 for the 2$^{nd}$ period, 0.36 for the 3$^{rd}$
period). These results suggest that the chemical and physical properties of aBBOA likely changed
much within the three periods despite similarities in PMF analysis. Both the primary gas pollutants

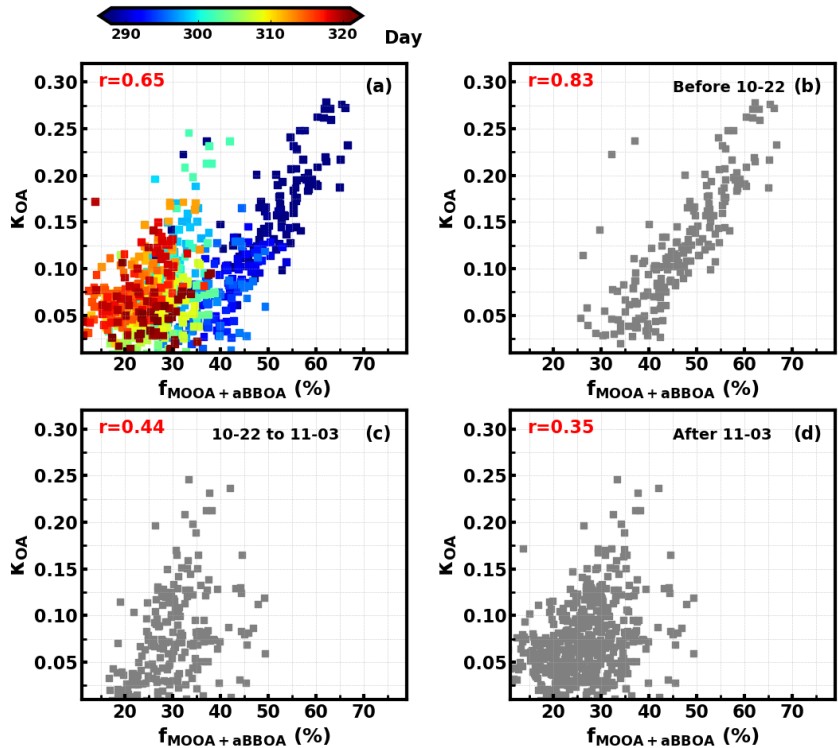

**Figure 6**. Relationships between $\kappa_{OA}$ and f$_{MOOA+aBBOA}$ during (a) the entire observation period; (b)10-12 to 10-22; (c) 10-23 to 11-02; (d) 11-03 to 11-17. Colors of scatter points in (a) represents day of the year.

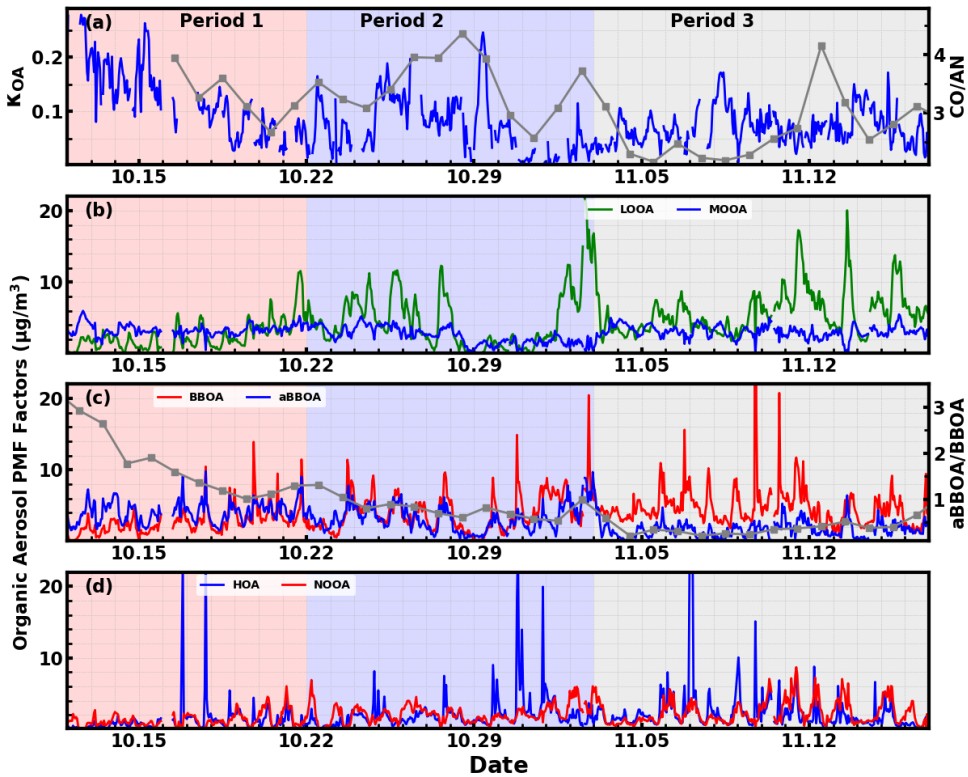

**Figure 7.** Time series of **(a)** derived $\kappa_{OA}$ and the right y-axis represent the ratio between CO and AN (acetonitrile); **(b)** LOOA and MOOA; **(c)** BBOA and aged BBOA, and the right axis represents the ratio between aBBOA/BBOA; **(d)** HOA and NOOA

CO and acetonitrile are highly associated with biomass burning and are often used as indicators of
biomass burning events, and the ratio between them can somehow indicate the emission profile
changes of biomass burning thus the primary VOC profile changes. The time series of the ratio between
CO and acetonitrile (Fig.7a) differs much during the three periods, especially for the second and the
third period. This difference suggests that although the biomass burning event continued, their
emission profiles associated with the burning fuels and conditions likely changed a lot, indicating that
aBBOA precursors might have changed during different agricultural activities, thus changing their
formation pathways as well as their chemical and physical properties. Other than the aBBOA property
changes, changes in OA factor contributions (for example, relative contributions of OA factors other
than MOOA and aBBOA) may also impact on the relationship between $\kappa_{OA}$ and $f_{\text{MOOA+aBBOA}}$. Also,
the chemical and physical properties of Night-OA and LOOA together with the VOC profile can also





433 have changed.

434  In general, the results shown here deliver the following key messages: (1) Although the O/C failed

435 to describe variations in $\kappa_{OA}$, variations of OA factors that are more related to VOC sources or OA

436 formation pathways could sometimes be found to explain the $\kappa_{OA}$ variations; (2) MOOA , being

437 highly oxygenated and associated with regional air mass, was the most important component that

438 enhanced $\kappa_{OA}$, which is consistent with current understandings, i.e., organic aerosol aging processes

439 have significant effects on $\kappa_{OA}$. However, the $\kappa_{OA}$ of secondary organic aerosol does not depends on

440 their O/C (contrary effects of aBBOA and LOOA on $\kappa_{OA}$); (3) Organic aerosol hygroscopicity of SOA

441 associated with similar sources might differ much under different conditions (effects of aBBOA on

442 $\kappa_{OA}$ differ much during different periods). These messages might be instructive to the

443 parameterization of $\kappa_{OA}$ in the following ways: (1) We might relate $\kappa_{OA}$ to VOC precursors in

444 laboratory studies, but the laboratory derived empirical relationship will likely fail in application of

445 ambient aerosols due to the formation pathway or the existence of other VOC precursors might result

446 in different chemical properties of ambient formed SOA, such as functional groups, from the laboratory

447 case; (2) It seems more plausible to find parameters other than O/C ratio to parameterize $\kappa_{OA}$, which

448 should be independent of sources and associated with the physical properties of OA. Overall, these

449 results further highlighted that $\kappa_{OA}$ parameterizations can be quite difficult and requires a lot of future

450 efforts.

451 **5 Conclusions**

452  In this study, a field campaign was conducted to characterize $\kappa_{OA}$ with high time resolution for

453 the first time at a rural site in the PRD region. The observation results showed that both typical $NO_3$

454 night chemistry (indicated by quick nighttime nitrate formation, extremely low NO concentration and

455 quick nighttime $O_3$ concentration decrease) and strong daytime photochemical chemistry (indicated

456 by high daytime $O_3$ concentration) prevailed during this field campaign. SOA dominated OA mass

457 (mass fraction >70% on average), which provided us a unique opportunity to investigate influences of

458 SOA formation on variations in organic aerosol hygroscopicity parameter $\kappa_{OA}$. Six OA factors were

459 resolved by the AMS PMF analysis, including two primary OA factors HOA and BBOA and other four

460 secondary OA factors MOOA, LOOA, aBBOA and Night-OA. The results demonstrated that mass

461 increase in both two primary OA factors had negative effects on $\kappa_{OA}$, which is consistent with current



understandings that POA components have quite low hygroscopicity (usually assumed as hydrophilic),
while SOA components had distinct effects on $\kappa_{OA}$. MOOA with the highest average O/C of 1 was
the most important factor that droves the increase of $\kappa_{OA}$, probably related with regional air mass and
local production contributes small. However, LOOA with average O/C slightly lower than that of
MOOA (O/C ~ 0.72), whose mass concentration increased dramatically during daytime due to local
production, had negative effects on $\kappa_{OA}$. Surprisingly, aBBOA with similar diurnal patterns to that of
LOOA, also formed quickly during daytime, but displayed much lower O/C (0.39), exerting positive
effects on $\kappa_{OA}$. In addition, $\kappa_{OA}$ revealed weak negative correlation to Night-OA fraction, which
increased during nighttime probably due to the $NO_3$ nighttime chemistry. This finding is in general
consistent with results of Suda et al. (2014) that the addition of $NO_3$ radical would exert negative
impacts on $\kappa_{OA}$. As a result, the contrasting effects of LOOA and aBBOA on $\kappa_{OA}$ resulted in the
weak correlation coefficient between $\kappa_{OA}$ and O/C. $\kappa_{OA}$ was mostly associated with mass fractions
of MOOA with highest O/C although its O/C is only a little higher than that of LOOA, which gave rise
to the weak but positive relationship between $\kappa_{OA}$ and O/C.

In general, the results presented in this study demonstrate that the O/C failed to describe variations

in $\kappa_{OA}$, however, SOA factors with different VOC sources or from different OA formation pathways
might have discrepant influences on the $\kappa_{OA}$. The contrasting effects of LOOA and aBBOA on $\kappa_{OA}$
demonstrated that VOC precursors from diverse sources and different SOA formation processes may
result in SOA with different chemical composition, functional properties as well as microphysical
structure, consequently influencing SOA hygroscopicity. On top of that, the hygroscopicity of SOA
associated with similar sources might also differ much during different emission and atmospheric
conditions. These results demonstrate that we might relate $\kappa_{OA}$ to VOC precursors in laboratory
studies, but the laboratory derived empirical relationships will likely fail in their application to ambient
aerosols due to the more complex formation pathways or the existence of other VOC precursors in the
ambient atmosphere, and thus difficult to apply in models. Overall, these results further highlighted
that $\kappa_{OA}$ parameterizations are quite complex, and it is important to conduct more researches on $\kappa_{OA}$
characterization under different meteorological and source conditions, and examine its relationship
with OA and VOC precursor profiles to reach a better characterization and come up with a more
appropriate parameterization approach for chemical and climate models.



**Data availability**. The data used in this study are available from the corresponding author upon request

Shan Huang (shanhuang_eci@jnu.edu.cn) and Min Shao (mshao@jnu.edu.cn)

**Competing interests**. The authors declare that they have no conflict of interest.

**Author Contributions**. YK and SH designed the aerosol experiments. YK conceived this research and wrote the manuscript together with SH. YK, BL and BX conducted aerosol light scattering enhancement factor measurements. QS, WC, WL, SH and WH conducted the SP-AMS measurements. MC, YK and SH conducted the particle number size distribution measurements. MS and BY planned this campaign. YP collected and managed criterial pollutants and meteorological parameters from Heshan supersite. PZ provided the humidified nephelometer system and contributed to discussions and revisions of the manuscript. DC and DY provided authority of conducting the campaign in Heshan supersite and gave data availability from the site. All other coauthors have contributed to this paper in different ways.

# Acknowledgments

This work is supported by the National Natural Science Foundation of China (grant No. 41805109, 41807302), National Key Research and Development Program of China (grant No. 2017YFC0212803, 2016YFC0202206, 2017YFB0503901), Key-Area Research and Development Program of Guangdong Province (grant No. 2019B110206001), Guangdong Natural Science Foundation (grant No. 2018A030313384), Special Fund Project for Science and Technology Innovation Strategy of Guangdong Province (grant No.2019B121205004), Guangdong Natural Science Funds for Distinguished Young Scholar (grant No. 2018B030306037) and Guangdong Innovative and Entrepreneurial Research Team Program (grant No. 2016ZT06N263).



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
