# Peer review of "Contrasting effects of secondary organic aerosol formations on organic aerosol"

_Atmospheric Chemistry and Physics, 2021_

## Referee Comment (RC1)

Ye Kuang et al. presented a field campaign results to characterize aerosol hygroscopicity with high time resolution at 80 % RH using humidified nephelometer. A high-resolution time-of-flight mass spectrometer was used to determine size-resolved aerosol chemical composition. Additionally, $NO_x$ and $O_3$ concentrations were measured, which allowed tracing diurnal variations of organic constituents. Using measurements results of particle size distribution as inputs of Mie theory and κ-Köhler model the hygroscopcicty $\kappa_{f,80\%}$ was obtained. Based on ZSR rule the organic hygroscopicity parameter, $\kappa_{OA}$ was retrieved and analysed. It was documented that $\kappa_{OA}$ weakly correlate with oxidation level parameter O/C of the total organic. More detailed analysis shown that $\kappa_{OA}$ negatively correlate with hydrophobic and night time formed OA fractions, while it positively correlate with aged biomass burning aerosol (aBBOA) (r = 0.35; O/C = 0.39) and more oxidized organic (MOOA) (r =0.64; O/C ~ 1). In contrast $\kappa_{OA}$ negatively correlate with low oxidized organic (LOOA) (r = -0.25) having O/C ratio of ~0.79. It is suggested that the contrasting effect of LOOA and aBBOA on $\kappa_{OA}$ is the result of a complex processes leading to SOA formation with different chemical composition, functional properties and microphysical structure, which are not captured by a single O/C parameter. The science in this paper is relevant to ACP's audience. The reviewer thinks that further clarification would be required prior to publication in ACP.

**General points:**

1. What is the aerosol resident time at 30 % and 80 % RH? Have you performed test measurements to make sure that the humidified residence time is sufficient to allow PM1 particles to grow to equilibrium? This is important, given that particles in the range of 200-800 nm make the main contribution to the scattering coefficient.

2. What is the accuracy in $\kappa_{OA}$ calculation? Besides environmental factors that are a mainly source of random errors, inaccuracy associated with the calculation of $\kappa_{f,RH}$, volume fractions and aerosol composition measurements can lead to serious systematic error. Have you estimated their input to $\kappa_{OA}$ when using Eq.(3)? If yes, please show it.

3. By default, it is assumed that at 80% humidity, the $\kappa_{f,RH}$, follows the ZSR rule and can be used to calculate $\kappa_{OA}$. Numerous studies have shown that at low RH, like 80% used in this study, the hygroscopic properties of multicomponent particles are not additive. Core-shell particle morphology, complex interaction between the components, limited solubility of sparingly soluble compounds, kinetic limitation caused by semi-solid state are the main factors, which lead to non-additive water uptake. These factors may provide significant uncertainty in $\kappa_{OA}$ . The CCN-retrieved $\kappa_{CCN}$ and $\kappa_{OA}$ may help to estimate the effect of low humidity on the uncertainty in scattering-derived $\kappa_{OA}$. Are the results of CCN measurements available for this field campaign?

   In my opinion, the $\kappa_{OA}$ and its uncertainty are poorly defined and additional efforts are needed to specify them. Otherwise, the conclusion about a weak relationship between kappa and O/C ratio and reasoning about the contrasting effects look unconvincing.

**Technical points**.

Line 232. Figure S2 have no panels 2a and 2b

Lines 257, 266 hydrophilic ?

Line 263. Fig.S4?

Line 282. NR-PM1 this abbreviation is not determined.

Line 335.   0.053± 0.006 ?   may be 0.53 ± 0.06

Line 346. There are no references to Fig. 4a-4f in the text, although all panels are labeled as (a)-(f).

Figure 5.  Blue line – NOOA. This  abbreviation is not determined. Why do need both panels?

Line 356. Fig. 3c?  May be Fig.5c.

Line 364.  Figure S4. Please check this figure. There is no nitrate concentration.

Figure 7.  Red line –NOOA. This abbreviation is not determined.

Supplement Fig.S4 . I did not find a text where this figure was discussed.

---

## Author Comment (AC1)

**Responses to anonymous referee #1**

**General Comment**:

Ye Kuang et al. presented a field campaign results to characterize aerosol hygroscopicity with high time resolution at 80 % RH using humidified nephelometer. A high-resolution time-of-flight mass spectrometer was used to determine size-resolved aerosol chemical composition. Additionally, NOx and O3 concentrations were measured, which allowed tracing diurnal variations of organic constituents. Using measurements results of particle size distribution as inputs of Mie theory and    -Köhler model the hygroscopicity $\kappa_{f,80\%}$ was obtained. Based on ZSR rule the organic hygroscopicity parameter, $\kappa OA$ was retrieved and analyzed. It was documented that $\kappa OA$ weakly correlate with oxidation level parameter O/C of the total organic. More detailed analysis shown that $\kappa OA$ negatively correlate with hydrophobic and night time formed OA fractions, while it positively correlate with aged biomass burning aerosol (aBBOA) (r = 0.35; O/C = 0.39) and more oxidized organic (MOOA) (r =0.64; O/C ~ 1). In contrast $\kappa OA$ negatively correlate with low oxidized organic (LOOA) (r = -0.25) having O/C ratio of ~0.79. It is suggested that the contrasting effect of LOOA and aBBOA on $\kappa OA$ is the result of a complex processes leading to SOA formation with different chemical composition, functional properties and microphysical structure, which are not captured by a single O/C parameter. The science in this paper is relevant to ACP's audience. The reviewer thinks that further clarification would be required prior to publication in ACP.

**Response**: We thank the reviewer for raising important questions that need to be clarified, which helped improve the quality and readability of this manuscript. Based on

the suggestion of the reviewer, we have performed several simulation experiments to quantify the accuracy of using $\kappa_{f(\mathrm{RH}),PM_{10}}$ represent $\kappa_{chem,PM_1}$ and quantify the uncertainties associated with $\kappa_{OA}$ derivation, estimate potential impacts of aerosol mass concentrations perturbations on the relationships between $\kappa_{OA}$ and organic aerosol factors. We believe that the revised manuscript is now more convincing than before.

**Major comments:**

**Comment**: What is the aerosol resident time at 30 % and 80 % RH? Have you performed test measurements to make sure that the humidified residence time is sufficient to allow PM1 particles to grow to equilibrium? This is important, given that particles in the range of 200-800 nm make the main contribution to the scattering coefficient.

**Response**: Thanks for your comment. The controlling factor that determines the aerosol equilibrium with the fixed RH environment is the sensing volume of nephelometer and the sample flow rate, and sensing volume of the nephelometer is about 0.4 L (from the brochure for Aurora 3000 nephelometer), and sample flow is 3 L/min, thus the estimated residence time of aerosol in the fixed RH environment is about 8 s. In addition, the residence time of aerosols in the humidifier tube and downstream tube is about 6 s. We did not perform the test measurements to make sure that the humidified residence time is sufficient to allow $PM_1$ particles to grow to equilibrium before the start of this field campaign, but previous experiences with the same instrument have shown that the

residence time is enough for equilibrium. This humidified nephelometer system is used for RH scan aerosol light scattering enhancement factor measurements with a flow rate of 4 L/min (larger than 3L/min of this campaign) (Zhao et al., 2019), the observed Ksca (RH=80%) show similar ranges with this field campaign. The results shown in Zhao et al. (2019) demonstrate that the equilibrium is always reached with RH>80%.

**Comment**: What is the accuracy in $\kappa OA$ calculation? Besides environmental factors that are a mainly source of random errors, inaccuracy associated with the calculation of $\kappa_{fRH}$ volume fractions and aerosol composition measurements can lead to serious systematic error. Have you estimated their input to $\kappa OA$ when using Eq.(3)? If yes, please show it.

**Response**: Thanks for your valuable suggestion. We agree with reviewer that we didn't perform comprehensive uncertainty analysis in the original manuscript, and we should show these results to boost readers' confidence about this method. In Eq.3, other than aerosol composition measurements, what's the accuracy of using $\kappa_{f(\text{RH}),PM_{10}}$ to represent $\kappa_{chem,PM_1}$ need to be carefully quantified, although simulation results using different PNSDs coupled with different size-resolved $\kappa$ distribution scenarios are discussed in Sect 3.2 of the original manuscript. As discussed in Kuang et al. (2020a), the difference between $\kappa_{chem,PM_1}$ and $\kappa_{f(\text{RH}),PM_{10}}$ are both influenced by aerosol PNSD and size-resolved $\kappa$ distribution. To cover as many cases as possible, we have performed a simulation to investigate the relative differences between $\kappa_{chem,PM_1}$ and $\kappa_{f(\text{RH}),PM_{10}}$ under different conditions of aerosol chemical compositions through varying size distributions of ammonium nitrate, ammonium sulfate, organic aerosol,

dust and BC, and their mass fractions randomly, and the mass size distributions follow lognormal distributions, that is:

$$dM/dlogDp = \frac{Mtot*fx}{\sqrt{2\pi}\log(\sigma_{g,x})}\exp\left[-\frac{(\log(D_p)-\log(D_{g,x}))^2}{2\log^2\sigma_{g,x}}\right],$$

Eq. 1

where x corresponding different aerosol compositions, and fx corresponding to its mass fractions. The parameters used in simulations are listed in Table 1, the number of randomly produced datasets for simulating $\kappa_{chem,PM_1}$ and $\kappa_{f(RH),PM_{10}}$ is 10000.

Table 1. used parameters for simulating differences between $\kappa_{chem,PM_1}$ and $\kappa_{f(RH),PM_{10}}$, m and $\sigma$ corresponds to average and standard deviation respectively.

| | fx range (%) | $D_g$ (nm) | | $\sigma_g$ | | $\rho$ | | $\kappa$ | |
|---|---|---|---|---|---|---|---|---|---|
| | | m | σ | m | σ | m | σ | m | σ |
| Ammonium Nitrate | 0-50 | 450 | 100 | 2.2 | 0.25 | 1.72 | | 0.56 | |
| Ammonium sulfate | 0-50 | 550 | 100 | 2.2 | 0.25 | 1.78 | | 0.56 | |
| Organic Aerosol | 10-60 | 450 | 150 | 2.2 | 0.25 | 1.2 | 0.05 | 0.15 | 0.05 |
| Dust | 5-50 | 2500 | 500 | 2.2 | 0.25 | 2 | | 0.05 | 0.015 |
| BC | 0-12 | 250 | 100 | 2.2 | 0.25 | 1.7 | | 0 | |

**Based on the simulation result, the first paragraph of the Sect3.2 is revised as the following**:

"The aerosol hygroscopicity parameter κ can be calculated from aerosol chemical composition measurements ($\kappa_{chem}$) on the basis of volume mixing rule, thus the organic aerosol hygroscopicity parameter $\kappa_{OA}$ were usually estimated through closure between measured κ and estimated κ using aerosol chemical measurements. In this study, the size-resolved aerosol chemical compositions of PM$_1$ were measured using the SP-AMS, however, the overall aerosol hygroscopicity was only derived based on aerosol light scattering measurements of PM$_{10}$ bulk aerosols. Results of Kuang et al. (2020a)

demonstrated that $\kappa_{chem}$ calculated based on bulk chemical compositions of PM$_1$ are quite consistent with $\kappa_{f(\text{RH})}$ of PM$_1$ ($\kappa_{chem,PM_1}$) therefore also consistent with $\kappa_{f(\text{RH})}$ of PM$_{10}$ ($\kappa_{f(\text{RH}),PM_{10}}$). However, simulation results in Kuang et al. (2020a) demonstrated that the ratio between $\kappa_{chem,PM_1}$ and $\kappa_{f(\text{RH}),PM_1}$ varies with PNSD and size-resolved κ distributions, and the applicability of this conclusion under varying aerosol chemical compositions and size distributions need further

[Figure]

**Figure 1**. Simulated $\kappa_{chem,PM_1}$ and $\kappa_{f(\text{RH}),PM_{10}}$, red texts give correlation coefficients, ratio=$\kappa_{f(\text{RH}),PM_{10}}/\kappa_{chem,PM_1}$, r$_{\text{eff}}$ is the effective radius of the aerosol populations, dashed red lines show the r$_{\text{eff}}$ range during the field campaign of this study.

clarification. Thus, we have designed a simulation experiments, to simulate the ratio between $\kappa_{chem,PM_1}$ and $\kappa_{f(\text{RH}),PM_{10}}$ considered wide ranges of aerosol chemical compositions and size distributions, details of the simulation are introduced in Part 2 of the supplement. The simulated results are shown in Fig.1. The results shows that the average relative difference ($\frac{\kappa_{f(\text{RH}),PM_{10}}-\kappa_{chem,PM_1}}{\kappa_{chem,PM_1}} \times$ 100%) was 2.1±5.3%, which demonstrates that in general $\kappa_{f(\text{RH}),PM_{10}}$ can be used to represent $\kappa_{chem,PM_1}$ under varying atmospheric conditions. The results also show that the ratio=$\kappa_{f(\text{RH}),PM_{10}}/\kappa_{chem,PM_1}$ is positively correlated with the effective radius of the aerosol population, which means that different levels of bias may exist under different PNSD conditions, and for effective radius range of this field campaign, the average relative difference is 0.7±4.9%. Given this, we have further simulated the $\kappa_{f(\text{RH})}$ of PM$_{10}$ and $\kappa_{chem}$ of PM$_1$ under different PNSDs of this campaign

coupled with different size-resolved κ distribution scenarios (as shown in Fig.S2a). As shown in the results in Fig.S2b, $\kappa_{chem,PM_1}$ and $\kappa_{f(RH),PM_{10}}$ are quite close to each other and the simulated average relative difference was -0.4±3%. Thus, $\kappa_{f(RH),PM_{10}}$ was used as the measured $\kappa_{chem,PM_1}$ in the following discussions."

Based on above analysis, 9% (3 standard deviations and based on results shown in Fig.S2b) can be used as the uncertainty of $\kappa_{chem,PM_1}$ to estimate the overall effects of parameter perturbations on $\kappa_{OA}$ derivations using Eq.3 during this field campaign. And the following part is added in Sect3.2 of the revised manuscript, after Eq.3 is introduced.

"Table 2. Effects of parameter perturbations on $\kappa_{OA}$ derivations using Eq.3

| Parameter | Uncertainty (3 standard deviations) | $\kappa_{OA}$ variations (1 standard deviation) |
|---|---|---|
| SO$_4$ mass concentration | 20% | 0.01 |
| NO$_3$ mass concentration | 20% | 0.006 |
| NH$_4$ mass concentration | 20% | 0.002 |
| OA mass concentration | 20% | 0.003 |
| $\kappa_{chem}$ | 9% | 0.014 |
| $V_{tot,PM1}$ | 25% | 0.003 |
| $\kappa_X$ | 0.03 | 0.003 |

The effects of $\kappa_{chem}$ perturbations, aerosol mass concentrations, $V_{tot,PM1}$ as well as $\kappa_X$ perturbations on $\kappa_{OA}$ derivations are simulated using Monte-Carlo method for each data point of the $\kappa_{OA}$ time series (1000 cases are randomly produced for each data point $\kappa_{OA}$) and average effects are summarized in Table 2. The perturbation parameter of $\kappa_{chem}$ is based on the simulation results using PNSDs of this field campaign shown in Fig.S2. The perturbation parameters of aerosol mass concentrations are consistent with Hong et al. (2018), and that of $V_{tot,PM1}$ is from Ma et al. (2011).

The perturbation parameter of $\kappa_X$ is specified based on that $\kappa$ of dust in general ranges from 0.01 to 0.08. The results show that the accuracy of using $\kappa_{f(\text{RH}),PM_{10}}$ to represent $\kappa_{chem,PM_1}$ affects most on $\kappa_{OA}$ derivations."

To test if effects of parameter perturbations on $\kappa_{OA}$ derivations have significant effects on the relationships between $\kappa_{OA}$ and organic aerosol PMF factors, we impose

[Figure]

**Figure 2**. Comparison between $\kappa_{OA}$ derived with and without random errors

random perturbations on parameters listed in Table 2 in each $\kappa_{OA}$ derivation. The comparison between originally derived $\kappa_{OA}$ and perturbed derivation of $\kappa_{OA}$ results is shown in Fig.4. The average difference between derived $\kappa_{OA}$ with and without random errors is 0, the standard deviation is 0.03. However, the relationships between $\kappa_{OA}$ derived with random errors and organic aerosol PMF factors changed only a little bit, and the results are shown in Fig.3. And these discussions are added in the revised manuscript as the following

"To test if effects of parameter perturbations on $\kappa_{OA}$ derivations have significant effects on the relationships between $\kappa_{OA}$ and organic aerosol PMF factors, we impose random perturbations on

parameters listed in Table 2 in each $\kappa_{OA}$ derivation. The comparison between originally derived $\kappa_{OA}$ and perturbed derivation of $\kappa_{OA}$ results is shown in Fig.S5. The average difference between derived

[Figure]

Figure 3. Correlations between $\kappa_{OA}$ derived with random errors and mass fractions of OA factors in total OA mass $\kappa_{OA}$ with and without random errors is 0, and the standard deviation is 0.03. However, the relationships between $\kappa_{OA}$ derived with random errors and organic aerosol PMF factors changed only a little bit, and the results are shown in Fig.S6."

**Comment**: By default, it is assumed that at 80% humidity, the $\kappa f, RH$, follows the ZSR

rule and can be used to calculate $\kappa OA$. Numerous studies have shown that at low RH, like 80% used in this study, the hygroscopic properties of multicomponent particles are not additive. Core-shell particle morphology, complex interaction between the components, limited solubility of sparingly soluble compounds, kinetic limitation caused by semi-solid state are the main factors, which lead to nonadditive water uptake. These factors may provide significant uncertainty in $\kappa OA$. The CCN retrieved $\kappa CCN$ and $\kappa OA$ may help to estimate the effect of low humidity on the uncertainty in scattering-derived $\kappa OA$. Are the results of CCN measurements available for this field campaign?

**Response**: Thanks for your suggestion. We agree with the reviewer that uncertainties embedded in the ZSR assumption might have significant impacts on $\kappa_{OA}$ derivation. But we might not have a better choice to investigate atmospheric $\kappa_{OA}$ evolutions based on field measurements. We thought that the derived $\kappa_{OA}$ might not be the real value of entire OA populations if only the OA populations are measured, however, the derived $\kappa_{OA}$ can be treated as ZSR-equivalent organic aerosol hygroscopicity, and how much $\kappa_{OA}$ derived with ZSR rule and the real one in atmospheric conditions needs further investigation, and we do not have a clue on this. Actually, at the very beginning, we also have similar worries with the reviewer because of the reasons the reviewer has mentioned. But we are curious about what can we get if the ZSR rule is used, just like what have been done by previous researches using field measurements HTDMA or CCN. We already got one successful try using the same method but aerosol chemical and light scattering enhancement factor (at RH of 85%) measurements are conducted in the North

China Plain (Kuang et al., 2020a). The final results have convinced us that derived $\kappa_{OA}$ values is meaningful for readers and the summarized conclusion should be convincing and new to the community: (1) The derived $\kappa_{OA}$ ranges are consistent with current literature values; (2)Most importantly, the $\kappa_{OA}$ derivations are completely independent of the PMF results, but we can get that derived $\kappa_{OA}$ of hourly time resolution is highly and positively correlated with that of MOOA; (3) The correlation coefficient between $\kappa_{OA}$ and $f_{MOOA+aBBOA}$ even reaches beyond 0.8 during certain period as shown in Fig.7. **Both experiences in the North China Plain and this campaign demonstrate that meaningful results can be obtained using ZSR rule under relatively lower RH conditions (80%-90%), thus we really want these results can promote more examinations about the applicability of ZSR rule in different RH and atmospheric relevant conditions**.

The reviewer mentioned that the CCN measurements might be helpful in constraining the $\kappa_{OA}$ uncertainty which is a good choice according to the physical understanding if the hygroscopicity of a particle is measured both with CCN instrument and the aerosol light scattering instrument. However, after careful thinking, we think the $\kappa_{CCN}$ can not be used to estimate influences of possible non-additive effects on $\kappa OA$ derivations. Size-resolved CCN measurements reflect overall hygroscopicity of aerosol population of diameter less than 200 nm that can be activated thus derived $\kappa_{OA}$ represents overall $\kappa$ of sub-200 nm organic aerosols, however, the $\kappa_{OA}$ derived using the method in this study represents the overall hygroscopicity of the entire organic aerosol population mainly ranges from 200-760 nm of $PM_1$ (Kuang et al., 2020a). Many

previous studies have demonstrated that chemical compositions of organic aerosol under different diameter ranges differ much, and are not the same at all (Kuang et al., 2020b), thus hinders the direct comparison between $\kappa_{OA}$ derived from light scattering enhancement measurements and CCN measurements.

**Comment**: In my opinion, the $\kappa OA$ and its uncertainty are poorly defined and additional efforts are needed to specify them. Otherwise, the conclusion about a weak relationship between kappa and O/C ratio and reasoning about the contrasting effects look unconvincing.

**Response**: We really appreciate your comments. The comments about the uncertainty of $\kappa_{OA}$ inspired us to perform a comprehensive evaluation about how much $\kappa_{chem,PM_1}$ and $\kappa_{f(\text{RH}),PM_{10}}$ differ under varying conditions of aerosol chemical compositions and size-distributions. And further quantified impacts of aerosol mass concentration perturbations and other key parameters on $\kappa_{OA}$ derivations.

**Technical points**.

**Comment**: Line 232. Figure S2 have no panels 2a and 2b

**Response**: Many thanks, we have added (a) and (b) in Fig.S2

**Comment**: Lines 257, 266 hydrophilic ?

**Response**: Thanks, changed to "hydrophobic"

**Comment**: Line 263. Fig.S4?Line 282. NR-PM1 this abbreviation is not determined.

**Response**: The average $\varepsilon_X$ is added in Fig.S4. And the NR-PM1 abbreviation is added

**Comment**: Line 335. 0.053± 0.006 ? may be 0.53 ± 0.06

**Response**: Thanks, should be 0.53 ± 0.06

**Comment**: Line 346. There are no references to Fig. 4a-4f in the text, although all panels are labeled as (a)-(f).

**Response**: Thank you, we have removed labels on these panels.

**Comment**: Figure 5. Blue line – NOOA. This abbreviation is not determined. Why do need both panels?

**Response**: The NOOA abbreviation is changed to Night-OA, and legend is changed on one panel.

**Comment**: Line 356. Fig. 3c? May be Fig.5c.

**Response**: Changed to Fig.5c

**Comment**: Line 364. Figure S4. Please check this figure. There is no nitrate concentration.

**Response**: Sorry for the typo of figure number which misled the reviewer. The sentence has been revised: "$\kappa_{OA}$ was also negatively correlated with Night-OA fraction, which

increased during nighttime (Fig. 6b). The Night-OA factor was highly correlated with nitrate concentrations (Figure S6), likely associated with the NO$_3$ nighttime chemistry as discussed in Sect. 4.1."

**Comment**: Figure 7. Red line –NOOA. This abbreviation is not determined. Supplement Fig.S4 . I did not find a text where this figure was discussed.

**Response**: The NOOA abbreviation is changed to Night-OA. The discussion about Fig.S4 is added in Sect3.2

Hong, J., Xu, H., Tan, H., Yin, C., Hao, L., Li, F., Cai, M., Deng, X., Wang, N., Su, H., Cheng, Y., Wang, L., Petäjä, T., and Kerminen, V. M.: Mixing state and particle hygroscopicity of organic-dominated aerosols over the Pearl River Delta region in China, Atmos. Chem. Phys., 18, 14079-14094, 10.5194/acp-18-14079-2018, 2018.

Kuang, Y., He, Y., Xu, W., Zhao, P., Cheng, Y., Zhao, G., Tao, J., Ma, N., Su, H., Zhang, Y., Sun, J., Cheng, P., Yang, W., Zhang, S., Wu, C., Sun, Y., and Zhao, C.: Distinct diurnal variation in organic aerosol hygroscopicity and its relationship with oxygenated organic aerosol, Atmos. Chem. Phys., 20, 865-880, 10.5194/acp-20-865-2020, 2020a.

Kuang, Y., Xu, W., Tao, J., Ma, N., Zhao, C., and Shao, M.: A Review on Laboratory Studies and Field Measurements of Atmospheric Organic Aerosol Hygroscopicity and Its Parameterization Based on Oxidation Levels, Current Pollution Reports, 10.1007/s40726-020-00164-2, 2020b.

Ma, N., Zhao, C. S., Nowak, A., Müller, T., Pfeifer, S., Cheng, Y. F., Deng, Z. Z., Liu, P. F., Xu, W. Y., Ran, L., Yan, P., Göbel, T., Hallbauer, E., Mildenberger, K., Henning, S., Yu, J., Chen, L. L., Zhou, X. J., Stratmann, F., and Wiedensohler, A.: Aerosol optical properties in the North China Plain during HaChi campaign: an in-situ optical closure study, Atmos. Chem. Phys., 11, 5959-5973, 10.5194/acp-11-5959-2011, 2011.

Zhao, P., Ding, J., Du, X., and Su, J.: High time-resolution measurement of light scattering hygroscopic growth factor in Beijing: A novel method for high relative humidity conditions, Atmospheric Environment, 215, 116912, 10.1016/j.atmosenv.2019.116912, 2019.

---

## Author Comment (AC2)

**Responses to anonymous referee #2**

**General comment**:

The manuscript of Kuang et al.: "Contrasting effects of secondary organic aerosol formations on organic aerosol hygroscopicity" shows that the oxidation state of the SOA does not always correlate to their degree of the hygroscopicity, and they propose that the hygroscopicity is controlled by additional factors along with the oxygenation degree of the OA. To show that, the authors use the data set of a field campaign conducted in China (Pearls River Delta) using a suite of online and high-resolution instrumentation. This paper is well written, and I recommend publication after the authors address comments below.

**Response**: We thank the reviewer for all the valuable comments and suggestions, which promote us to learn from the book *Atmospheric Chemistry and Physics: From Air Pollution to Climate Chang* and provide more solid calculations to improve our narrative.

**Major Comments:**

**Comment**: Lines 166-171: The authors state that they did not perform a calibration at the sampling state. They used an RIE=4 for ammonium taken from the last calibration. This could be a serious issue for ammonium concentration resulting in artifacts (lower of higher ammonium concentration). How well the ammonium concentration correlated with filter measurements (mentioned on line 178) using a RIE=4? Please provide the

R2. Also please explain how stable this parameter for the specific instrument is. How much does the ammonium concentration changes if you use a RIE of 3.0, 3.5 and 4.5? Please provide the corresponding NH4 mass concentrations and its % contribution to the PM1 for the above 3 cases.

**Response:** Thanks for reviewer's comment. Actually, we have similar worries about the data accuracy of SP-AMS measurements at the very beginning, so we paid a special attention to assure the data quality. Instrument internal calibrations such as m/z calibration and single ion signal scan were performed every day and offline samples were collected daily during the campaign. Aerosol chemical properties measured by parallel instruments were also collected to validate the SP-AMS data. As mentioned in the manuscript, excellent agreement between particle volume concentration derived from SP-AMS and SMPS may support the data accuracy of SP-AMS regarding the total volume (mass) concentration (shown in Figure S5). For better showing the data accuracy, we added Figure S7, S8 and S9 to present the comparisons between the data from SP-AMS and PM$_{2.5}$ monitoring device, as well as GAC, ECOC analyzer, AE33 aethalometer and offline measurements. The good correlation (slope =0.74, R$^2$ = 0.80) between the total **PM$_1$** mass concentration from SP-AMS (BC included) and **PM$_{2.5}$** mass concentration indicates SP-AMS provided correct total mass concentration. The comparisons of individual components between SP-AMS and external measurements (Figure S8 and S9) also showed good agreements and consequently assure the data quality. Specifically, ammonium (NH$_4$) mass concentration from SP-AMS was well correlated with offline result (with PM$_{2.5}$ impactor), with a reasonable slope of 0.76 and

a good correlation coefficient (R= 0.83, $R^2$ = 0.69). As reviewer suggested, we used RIE of 3, 3.5 and 4.5 to calculate $NH_4$ mass concentration as well as its contribution to $PM_1$. As shown in Table 1, $NH_4$ mass concentrations varied from 2.99 (RIE=4.5) to 4.48 μg/$m^3$ (RIE = 3) with tiny difference on mass fractions (8.18%~11.76%, while 9.10% for RIE = 4 was considered in previous analysis). Moreover, comparisons to offline results (Figure 1) also suggest RIE of 4 could be a good choice because applying RIE of 3 and 3.5 might overestimate the $NH_4$ concentration (slope = 1.01 and 0.86) while using RIE =4.5 might underestimate it (slope = 0.67) considering total $PM_1$ mass concentration (derived from SP-AMS) took 74% of total $PM_{2.5}$ mass concentration (Figure S5).

**Table 1** $NH_4$ mass concentration and its contribution to $PM_1$ with different RIE

| RIE | Average concentration (μg/$m^3$) | Mass fraction in PM1 (%) |
|---|---|---|
| **RIE=4** | 3.39 ± 1.54 | 9.10% |
| **RIE=3** | 4.48 ± 2.07 | 11.76% |
| **RIE=3.5** | 3.84 ± 1.77 | 10.26% |
| **RIE=4.5** | 2.99 ± 1.38 | 8.18% |

[Figure]

**Figure 1** Comparisons of NH$_4$ mass concentration between SP-AMS and offline measurements (PM$_{2.5}$) with different RIEs of NH$_4$.

**Comment**: Lines 235-243, Table 1 and equation 2: How it is possible to have (NH4)HSO4 and (NH4)2SO4 and NH4NO3 at the same time? Ammonia will first neutralize all available sulfate and it will bring the aerosol in the form of (NH4)2SO4. Then, whatever ammonia exists in the atmosphere will react with HNO3 to form NH4NO3. The co-existence of (NH4)HSO4 and NH4NO3 is not compatible according to thermodynamic lows. (Atmospheric Chemistry and Physics: From Air Pollution to Climate Change, Seinfeld and Pandis, Wiley, 3rd edition, 2016). The mass balance should be recalculated. After that, please discuss the potential compounds present in the

atmosphere during the campaign.

(NH4)HSO4 should be deleted from equation (2) and (3) and the calculations should be done again. How much the results (kOA) do change with this correction?

**Response**: Thanks for your comment, we think the reviewer raised a very interesting question. Actually we didn't think of this at all when we were using the ion pairing scheme proposed by Gysel et al. (2007) which have been used in many aerosol hygroscopicity closure or organic aerosol hygroscopicity estimation studies since then.

We have read the book *Atmospheric Chemistry and Physics: From Air Pollution to Climate Chang* and find discussions about the Ammonia-Nitric Acid-Sulfuric Acid-Water system in Chapter 10.4.4, and we also find the figure 10.23 which shows there is no way that both $HSO_4^-$ and $NO_3^-$ can be existed at the same time. The calculation in chapter 10.4.4 follows the way that Ammonia will first neutralize all available sulfate and it will bring the aerosol in the form of $(NH_4)_2SO_4$, and the free part ammonia will then react with $HNO_3$ to form $NH_4NO_3$. However, we think this a simplified calculation, and this conclusion does not hold in the real atmosphere (if completely holds, no nitrate should be exist in ammonia poor conditions, which is not the case).

Actually, another reason we never realized this is that we believe that **an equilibrium will always exist between $HSO_4^-$ and $SO_4^{2-}$ under acid conditions through $HSO_4^- \leftrightarrow H^+ + SO_4^{2-}$, and ammonia concentrations under general atmospheric conditions does not raise aerosol pH to exceed pH=7** (Guo et al., 2017), and multiphase buffering mechanism must be considered in considering aerosol pH and ionic form of sulfate (Zheng et al., 2020). If all sulfate is in the form of $(NH_4)_2SO_4$, then

the aerosol should not be acidic. We have initiated a simulation using a thermodynamic model ISORROPIA thus followed both the thermodynamic rule and also the partitioning theory of volatile gases like ammonia and nitric acid. Case 1: total ammonia concentration is 17 ug/m3 (1 umol/m$^3$), sulfate is 19.2 ug/m3 (0.2 umol/m$^3$) and nitrate is zero, which means that the ammonia concentration exceeds far the required ammonia for sulfate neutralization, and the RH is 80%, T is 298 K. The simulated aqueous fraction of $HSO_4^-$ in total sulfate is 0.03 umol/m$^3$ which means that even under this condition, $HSO_4^-$ exists and cannot be neglected. Case 2: change nitrate to 6.3 ug/m3 (0.1 umol/m$^3$) and other parameters are same with Case 1, thus still ammonia rich condition. For case 2, the simulated $HSO_4^-$ is 0.013 umol/m$^3$. Case 3: reduce total ammonia concentration is 8.5 ug/m$^3$ (0.5 umol/m$^3$), and other parameters remain the same with Case 2. For Case 3, the simulated $HSO_4^-$ will be 0.1 umol/m$^3$, which means that 50% of sulfate is in the form of $HSO_4^-$.

These simulation results demonstrate that $HSO_4^-$ , $SO_4^{2-}$ and $NO_3^-$ coexist in the aqueous phase. Due to the lack of gas phase ammonia and nitric acid mass concentration measurements, we didn't perform simulations using thermodynamic model but still use the widely used ion pairing scheme proposed by Gysel et al. (2007), and thus keep same way of estimating $\kappa_{OA}$ with previous studies for the comparison convenience.

**Comment:** Lines 293-294: Is this NH4NO3 formation during the night due to the lower temperature during the night? The dissociation constant Kp(T) of the ammonia-nitic acid system (NH3(g) + HNO3 (g) ↔ NH4NO3 (s)) is a function of the temperature and

it is very sensitive to the temperature changes. As the temperature decreases Kp(T) also decreases, and the equilibrium shifts towards the aerosol phase increasing the NH4NO4 mass concentration (Atmospheric Chemistry and Physics: From Air Pollution to Climate Change, Seinfeld and Pandis, Wiley, 3rd edition, 2016). It seems that this is the case taking account Figures 2a and 2c. Please discuss.

**Response**: Thanks for your comment. After reading your comments, we realized that we should be more cautious when discussing the possible nitrate formation mechanism. You mentioned a very good point that the increase of nitrate might be a result of solid and gas phase equilibrium transition due to the temperature decrease. Depending on the RH or the RH history that aerosol particles have been experienced, the aerosol particles might be solid (if crystallized) or aqueous phase (dehydration branch from the morning to the afternoon, metastable) since about 16:00. However, when aerosol particles will be deliquescent is also a puzzle if aerosol particle crystallized in the afternoon due to the complex dependence of deliquescence RH on aerosol mixtures of ammonium sulfate, ammonium nitrate and organic aerosols (Kuang et al., 2016), and the peak RH in the morning was usually lower than 80%,  To illustrate the possible mechanisms behind the nitrate increase. We assume the following two states of ammonium nitrate: (1) solid

state; (2) aqueous phase.

For the solid phase case, the $NH_3(g) + HNO_3(g) \leftrightarrow NH_4NO_3(s)$ equilibrium theory is applied (Seinfeld and Pandis, 2016). If we have the initial concentration of gaseous $NH_3$, then we can guess the initial $HNO_3(g)$ based on the $K_p(T)$ and estimate the nitrate concentration increase due to the decrease $K_p(T)$. The $K_p(T)$ changed from 84 $ppb^2$ to 24 $ppb^2$ from 16:30 to 23:30 for the average case with air temperature decrease from 27.7 to 22.7 °C. Figure 2 shows the variations of nitrate mass concentrations under

[Figure]

**Figure 2**. Simulated variations of nitrate under different $NH_3$ (g) conditions by assuming aerosol particles are aqueous using the ISORROPA thermodynamic model.

different assumed $NH_3$ (g) concentration conditions by considering variations of $K_p(T)$. The results demonstrate that the temperature induced $K_p(T)$ change is enough for explaining observed nitrate increase, and thus a possible mechanism.

For the aqueous phase case, $NH_4NO_3$ will be found in the aqueous aerosol phase, and the corresponding dissociation reaction is then $NH_3(g) + HNO_3(g) \leftrightarrow NH_4^+ + NO_3^-$. For this case, both temperature decrease and RH increase play roles in the $NH_3$ and $HNO_3$ partitioning. We use a thermodynamic model ISORROPIA (forward, metastable) to simulate the possible magnitude of nitrate increase driven by increased RH and reduced temperature for the average case shown in Fig.2 of the manuscript. We also assume different scenarios of $NH_3(g)$ concentrations and for each $NH_3(g)$ concentration scenario, we estimate the $HNO_3(g)$ concentration with which we can reproduce the observed nitrate concentration at the initial time (16:30), to get a total $HNO_3$ concentration in gas and aerosol phase, then we can estimate the possible increase of nitrate in aerosol phase due to repartitioning and the results are shown in Fig.3. The results show that the more $HNO_3$ dissolved in aqueous phase can also explain the observed nitrate mass concentration increase.

The simulation results shown above demonstrate that repartitioning of $HNO_3$ (total $HNO_3$ in aerosol and gas phase remain unchanged) from gas phase to aerosol phase can explain the observed nitrate mass concentration increase under both assumptions that aerosols are completely aqueous or solid. The actual atmospheric case is more likely that both solid (freshly emitted and crystallized) and aqueous particles (dehydration branch, not crystallized) exist, however, does not affect conclusion that the repartitioning of $HNO_3$ in gas and aerosol phase can explain the observed nitrate increase. In addition, NO concentration is quite low (almost zero) after 16:00 which might be favorable of $N_2O_5$ formation and the $N_2O_5$ hydrolysis as a possible nitrate

formation pathway cannot be excluded. Thus, texts in the manuscript which are related to the delineation of nitrate formation are revised accordingly. For example, the second paragraph of Sect 4.1 is revised as:

"The average diurnal variations of $NO_2$, NO, $O_3$, CO, aerosol chemical compositions, $\kappa_{f(RH)}$ and meteorological parameters are shown in Fig.2. O3 concentrations began to increase after sunrise, peaked near 15:00 and then began to decrease quickly but drops slower after midnight. Meanwhile, NO concentration began to decrease quickly after sunrise, reached and remained near zero after noontime, and began to slightly increase after 21:00. $NO_2$ concentration increased quickly after 15:00 and reached a plateau after 21:00. Variation characteristics of NO, $O_3$, and $NO_2$ suggest that the relatively low NO concentration resulted in weak titration effects on $O_3$, whereupon typical $NO_3$ chemistry and subsequent $N_2O_5$ chemistry might occur, which might contribute to the observed nitrate increase after sunset. However, nitrate concentrations increased quickly after about 16:00 and peaked after midnight (about 03:00 LT), indicating that there must be a mechanism is responsible for the observed nitrate increase at least before sunset. To dig more into this, the possible pathways of nitrate formation since 16:00 was simulated and discussed in Sect.3 of the supplement. The results demonstrate that the repartitioning of $HNO_3$ in gas and aerosol phase due to the temperature decrease and RH increase can mainly explain the observed nitrate increase. And the strong daytime photochemistry and decrease of $NO_2$ concentration might result in significant production of gas phase before about 16:00. However, the possible contribution of $N_2O_5$ hydrolysis to nitrate formation cannot be excluded."

**Comment**: Lines 307-310: This explanation is not sufficient stated. If this is the case (i.e., N2O5 production during the night, then NO2 should be reduced, but Figure 2f shows that NO2 increases during the night. Please make sure that you explain the phenomena correctly.

**Response**: Thanks for your comment, but we cannot agree with the reviewer on this. The $N_2O_5$ production does not require that $NO_2$ should be reduced. The variation of $NO_2$ mass concentration depends on its sources and sinks, if its source is relatively stronger then the $N_2O_5$ production only slows down the $NO_2$ concentration increase instead of reducing $NO_2$ mass concentration. The main factor that drives the speculation that $NO_3$ and $N_2O_5$ will be produced is that during night the $O_3$ concentration is still higher than 25 ppb before 24:00, however the NO concentration is almost zero since 16:00.

In addition, we have analyzed the possible mechanism behind the observed nitrate increase after 16:00 as suggested by the reviewer in the previous comment, and have revised this part as the following:

"The average diurnal variations of $NO_2$, NO, $O_3$, CO, aerosol chemical compositions, $\kappa_{f(RH)}$ and meteorological parameters are shown in Fig.2. O3 concentrations began to increase after sunrise, peaked near 15:00 and then began to decrease quickly but drops slower after midnight. Meanwhile, NO concentration began to decrease quickly after sunrise, reached and remained near zero after noontime, and began to slightly increase after 21:00. $NO_2$ concentration increased quickly after 15:00 and reached a plateau after 21:00. Variation characteristics of NO, $O_3$, and $NO_2$ suggest

that the relatively low NO concentration resulted in weak titration effects on $O_3$, where upon typical $NO_3$ chemistry and subsequent $N_2O_5$ chemistry might occur, which might contribute to the observed nitrate increase after sunset. However, nitrate concentrations increased quickly after about 16:00 and peaked after midnight (about 03:00 LT), indicating that there must be a mechanism is responsible for the observed nitrate increase at least before sunset. To dig more into this, the possible pathways of nitrate formation since 16:00 was simulated and discussed in Sect.3 of the supplement. The results demonstrate that the repartitioning of $HNO_3$ in gas and aerosol phase due to the temperature decrease and RH increase can mainly explain the observed nitrate increase. And the strong daytime photochemistry and decrease of $NO_2$ concentration might result in significant production of gas phase before about 16:00. However, the possible contribution of $N_2O_5$ hydrolysis to nitrate formation cannot be excluded."

**Comment**: Line 310: "Nitrate concentrations increased quickly since 16:00:". This is contradictory to Figure 2c, where nitrate decreases from 09:00 to 16:00 and then starts increasing, reaching its maximum at 03:00.

**Response**: Thanks, we have revised this sentence as "However, nitrate concentrations increased quickly after about 16:00 and peaked after midnight (about 03:00 LT), indicating that there must be a mechanism is responsible for the observed nitrate increase at least before sunset."

**Comment**: Line 312: "nighttime heterogenous formation of nitrate, " please check if

this is the case.

**Response**: we rephrased this sentence as "Under the strong daytime photochemistry and nighttime increase of nitrate"

**Comment**: Line 337: "On average, $\delta$ œ¿$\delta$ '¶$\delta$ '¨ increased slowly during the nighttime". From Figure 3c it seems that kOA is rather stable during the night and it increases after 03:00 (a.m.).

**Response**: It can be seen from Fig.3c, that $\kappa_{OA}$ increased slowly since 18:00 and remain stable during 0 to 3:00 and then increase.

**Comment**: Lines 357-359: "However, $\kappa_{OA}$ was also negatively correlated with LOOA 357 (Fig.4d), whose mass concentration increase rapidly after sunrise and are likely secondary due to local photochemistry with potential precursors such as isoprene and anthropogenic VOCs." How do you support this? Could you discuss any results from the gas-phase?

**Response**: The observation site is on a small mountain of Heshan county, is about 55 km away from megacity Guangzhou and is surrounded by villages and small residential towns, and the surrounding areas are covered with trees as shown in Fig.S1. Thus, the VOC precursors of observed quick LOOA formation are likely both biogenic and anthropogenic. Though we do not have detailed results about VOCs at this moment, a similar diurnal pattern of LOOA and ozone (Figure 3) may indicate that these two secondary products were formed in local photochemical process with VOC precursors

emitted from surrounding area. So, we rephrased this sentence as "whose mass concentration increase rapidly after sunrise and are likely secondary due to local photochemistry with potential precursors such as isoprene of both biogenic and anthropogenic VOCs as the observation site is surrounded by small towns and areas with high percentage cover of trees as shown in Fig.S1"

[Figure]

Figure 4. Diurnal patterns of $O_3$ and LOOA during the campaign.

**Minor Comments**:

**Comment**: Please use past tense throughout the whole manuscript. There are parts that the present tense alternates with past tense (e.g., section 2.2).

**Response**: Thanks, we went through the manuscript and used past tense throughout the whole manuscript.

**Comment**: Line 62: Please replace "evolvement" with "evolution".

**Response**: changed accordingly.

**Comment**: Lines 69-71: Please add here that the volatility could be another factor that affects the hygroscopicity of the SOA.

**Response**: We did not add volatility because volatility itself is also determined by carbon-chain length, functional groups, etc. Both volatility and hygroscopicity are physical properties of organic aerosol and determined by organic aerosol structure, they may be related, but the hygroscopicity is not essentially affected by volatility.

However, the reviewer raised a good point about organic aerosol volatility and hygroscopicity, thus we have added this point in Sect 4.3 as introduced in the response of the next comment.

**Comment:** Lines 71-72: Please cite here the following 4 papers:

Kuwata, M., Kondo, Y., Mochida, M., Takegawa, N., and Kawamura, K.: Dependence of CCN activity of less volatile particles on the amount of coating observed in Tokyo, J. Geophys. Res., 112, D11207, doi:10.1029/2006JD007758, 2007.

Asa-Awuku, A., Engelhart, G. J., Lee, B. H., Pandis, S. N., and Nenes, A.: Relating CCN activity, volatility, and droplet growth kinetics β-caryophyllene secondary organic aerosol, Atmos. Chem. Phys., 9, 795–812, 2009.

Frosch, M., Bilde, M., Nenes, A., Praplan, A. P., Jurányi, Z., Dommen, J., Gysel, M., Weingartner, E., and Baltensperger, U.: CCN activity and volatility of β-caryophyllene secondary organic aerosol, Atmos. Chem. Phys., 13, 2283–2297, 2013.

Kostenidou, E., Karnezi, E., Hite Jr., J. R., Bougiatioti, A., Cerully, K., Xu, L., Ng, N. L., Nenes, A., and Pandis, S. N.: Organic aerosol in the summertime southeastern United States: components and their link to volatility distribution, oxidation state and hygroscopicity, Atmos. Chem. Phys., 18, 5799–5819, 2018.

**Response**: Thanks, the reviewer mentioned several important papers about organic aerosol hygroscopicity and volatility. We did not cite these papers in Lines 71-72, but we have cited these papers in Sect 4.3 as the following "It seems more plausible to find parameters other than O/C ratio to parameterize $\kappa_{OA}$, which should be independent of sources and associated with the physical properties of OA, such as volatility (Kuwata et al., 2007;Asa-Awuku et al., 2009;Frosch et al., 2013;Kostenidou et al., 2018)"

**Comment**: Lines 106-110: Please rewrite this part. It should be not mentioned any sections, but rather describe and explain what it will follow in the next.

**Response**: Thanks, we have revised this part as " We described details on aerosol measurements and the $\kappa_{OA}$ estimation method in measurements and method part. In the results and discussion section, we first sketched out the overview of campaign measurements and then discussed the $\kappa_{OA}$ variation characteristics as well as its influencing factors, and in the last part, the complexity regarding $\kappa_{OA}$ parameterization was further demonstrated and elucidated. The summaries are provided in the conclusion part."

**Comment**: Line 116: Please replace "locates" with "was located".

**Response**: changed accordingly.

**Comment**: Line 117: Please add "the" before "megacity".

**Response**: added accordingly.

**Comment**: Line 130: Please add "the" before "physical".

**Response**: added accordingly.

**Comment:** Line 132: Please add "the" before "aerosol".

**Response**: added accordingly.

**Comment**: Line 134: Please replace "of" with "the".

**Response**: replaced accordingly.

**Comment**: Line 135: Please add "the" before "aerosol".

**Response**: added accordingly.

**Comment**: Lines 137-139: Please rephrase this sentence.

**Response**: This sentence is rephrased as "To make sure the accuracy of the measured RH in the sensing volume of the wet Nephelometer, three Vaisala HMP110 sensors with accuracies of ($\pm 0.2$ ℃ and $\pm 1.7$ % for RH between 0 to 90%) were used to monitor the RH at different parts of the wet nephelometer."

**Comment**: Line 139: Please replace "Two" with "Two sensors".

**Response**: added accordingly.

**Comment**: Line 146: Please add "a" before "flow".

**Response**: added accordingly.

**Comment**: Line 146: Please add "the" before "sampling".

**Response**: added accordingly.

**Comment**: Line 149: Please add "the" before "particle".

**Response**: added accordingly.

**Comment:** Line 152: Please add "the" before "size-resolved".

**Response**: added accordingly.

**Comment:** Line 153: Please delete "basically".

**Response**: deleted accordingly.

**Comment**: Lines 161-162: Please rephrase this sentence.

**Response**: This sentence is rephrased as "The air flow in the AMS was first controlled by the orifice and then focused through the aerodynamic lens of SP-AMS, and then

particles with diameter in sub-micrometer range were detected."

**Comment**: Line 183: "As a wildly used source analysis method" I am not sure what do you mean here.

**Response**: we deleted those words.

**Comment**: Lines 186-188: Please rephrase this sentence.

**Response**: Thanks, the sentence is rephrased as "In this study, PMF using high resolution AMS data including two matrices (organic ion mass concentrations and their uncertainties) were conducted by an Igor Pro-based panel, i.e., PMF Evaluation Tool (PET, v2.06, Ulbrich et al., 2009), following the instruction in Ulbrich et al. (2009)."

**Comment**: Lines 190-199: This is not the right place for this paragraph. It should be moved to the Results section.

**Response**: Thanks, we think this part should be in the measurements part, because this is the description of how PMF factors are determined.

**Comment**: Lines 206-207: Please rephrase this sentence.

**Response**: Thanks, this sentence is rephrased as "and $\kappa_{f(\mathrm{RH})}$ represents a diameter independent hygroscopicity parameter $\kappa$ that fits the observed $f(80\%, 525\,\mathrm{nm})$ best and solved through iteration algorithm."

**Comment**: Lines 223-226: Please rephrase this sentence.

**Response**: Thanks, this sentence is rephrased as "The aerosol hygroscopicity parameter $\kappa$ can be calculated from aerosol chemical composition measurements ($\kappa_{chem}$) on the basis of volume mixing rule, thus the organic aerosol hygroscopicity parameter $\kappa_{OA}$ were usually estimated through closure between measured $\kappa$ and estimated $\kappa$ using aerosol chemical measurements."

**Comment**: Line 250: Please define better the parameters $\kappa$ and $\varepsilon$.

**Response**: Thanks, the sentence is rephrased as "Where $\kappa_i$ is hygroscopicity parameter $\kappa$ of compound i , and $\varepsilon_i$ is volume fraction of compound i in the mixture (Vi/Vtot, Vi and Vtot are volume of compound i and total aerosol volume of $PM_1$ )."

**Comment**: Lines 255-256: Please rephrase this sentence.

**Response**: Thanks, this sentence is rephrased as "These unidentified aerosol species, in continental regions, likely be dust but still possible composed of other components such as biogenic primary aerosol."

**Comment**: Line 352: Please provide the corresponding literature.

**Response**: Reference is added.

**Comment**: Line 356: "It was generally thought that secondary aerosol formation would result…", This phrase is not well connected to the previous sentences.

**Response**: This sentence is put after the sentence that originally after it, and thus revised as "However, $\kappa_{OA}$ was also negatively correlated with LOOA (Fig.4d), whose mass concentration increase rapidly after sunrise and are likely secondary due to local photochemistry with potential precursors of both biogenic and anthropogenic VOCs as the observation site is surrounded by small towns and areas with high percentage cover of trees as shown in Fig.S1. The negative correlation between $\kappa_{OA}$ and LOOA is contradictory with the generally thought that secondary aerosol formation would result in increases of aerosol hygroscopicity."

**Comment**: Line 359: What do you mean by average O/C"? Each factor derived from PMF analysis has a constant O/C ratio.

**Response**: The word "average" is deleted.

**Comment**: Lines 360-363: This sentence is quite big and complicate. Please rephrase and simplify.

**Response**: Thanks, this sentence is revised as "The negative correlation between $\kappa_{OA}$ and LOOA mass fraction explained why O/C failed to describe diurnal variations of $\kappa_{OA}$: the O/C ratio for LOOA is 0.72, which is only lower than that of MOOA, suggesting that the daytime LOOA formation and decrease of BBOA and HOA mass concentrations drove the increase of daytime O/C but the $\kappa_{OA}$ didn't follow."

**Comment**: Line 394: Please replace "?" with ".".

**Response**: replaced.

Asa-Awuku, A., Engelhart, G. J., Lee, B. H., Pandis, S. N., and Nenes, A.: Relating CCN activity, volatility, and droplet growth kinetics of β-caryophyllene secondary organic aerosol, Atmos. Chem. Phys., 9, 795-812, 10.5194/acp-9-795-2009, 2009.

Frosch, M., Bilde, M., Nenes, A., Praplan, A. P., Jurányi, Z., Dommen, J., Gysel, M., Weingartner, E., and Baltensperger, U.: CCN activity and volatility of β-caryophyllene secondary organic aerosol, Atmos. Chem. Phys., 13, 2283-2297, 10.5194/acp-13-2283-2013, 2013.

Guo, H., Weber, R. J., and Nenes, A.: High levels of ammonia do not raise fine particle pH sufficiently to yield nitrogen oxide-dominated sulfate production, Scientific reports, 7, 12109, 10.1038/s41598-017-11704-0, 2017.

Gysel, M., Crosier, J., Topping, D. O., Whitehead, J. D., Bower, K. N., Cubison, M. J., Williams, P. I., Flynn, M. J., McFiggans, G. B., and Coe, H.: Closure study between chemical composition and hygroscopic growth of aerosol particles during TORCH2, Atmos. Chem. Phys., 7, 6131-6144, 10.5194/acp-7-6131-2007, 2007.

Kostenidou, E., Karnezi, E., Hite Jr, J. R., Bougiatioti, A., Cerully, K., Xu, L., Ng, N. L., Nenes, A., and Pandis, S. N.: Organic aerosol in the summertime southeastern United States: components and their link to volatility distribution, oxidation state and hygroscopicity, Atmos. Chem. Phys., 18, 5799-5819, 10.5194/acp-18-5799-2018, 2018.

Kuang, Y., Zhao, C. S., Ma, N., Liu, H. J., Bian, Y. X., Tao, J. C., and Hu, M.: Deliquescent phenomena of ambient aerosols on the North China Plain, Geophysical Research Letters, n/a-n/a, 10.1002/2016GL070273, 2016.

Kuwata, M., Kondo, Y., Mochida, M., Takegawa, N., and Kawamura, K.: Dependence of CCN activity of less volatile particles on the amount of coating observed in Tokyo, Journal of Geophysical Research: Atmospheres, 112, https://doi.org/10.1029/2006JD007758, 2007.

Seinfeld, J., and Pandis, S.: Atmospheric Chemistry and Physics: From Air Pollution to Climate Change, Third Edition, 2016.

Zheng, G., Su, H., Wang, S., Andreae, M. O., Pöschl, U., and Cheng, Y.: Multiphase buffer theory explains contrasts in atmospheric aerosol acidity, Science, 369, 1374, 10.1126/science.aba3719, 2020.